**Phytoplankton growth and consumption by microzooplankton**
**stimulated by turbulent nitrate flux suggest rapid trophic transfer**
**in the oligotrophic Kuroshio**
Toru Kobari[1*], Taiga Honma[2], Daisuke Hasegawa[3], Naoki Yoshie[4], Eisuke Tsutsumi[5], Takeshi
Matsuno[6], Takeyoshi Nagai[7], Takeru Kanayama[2], Fukutaro Karu[2], Koji Suzuki[8], Takahiro Tanaka[3],
Xinyu Guo[4], Gen Kume[1], Ayako Nishina[1] and Hirohiko Nakamura[1]
[1]Aquatic Sciences, Faculty of Fisheries, Kagoshima University
4-50-20 Shimoarata, Kagoshima, Kagoshima 890-0056, Japan
[2]Aquatic Sciences, Graduate School of Fisheries, Kagoshima University
4-50-20 Shimoarata, Kagoshima, Kagoshima 890-0056, Japan
[3]Tohoku National Fisheries Research Institute, Japan Fisheries Research and Education Agency
3-27-5 Shinhama-cho, Shiogama, Miyagi 985-0001, Japan
[4]Center for Marine Environmental Studies, Ehime University
2-5 Bunkyo-cho, Matsuyama, Ehime 790-8577, Japan
[5]Atmosphere and Ocean Research Institute, University of Tokyo
5-1-5 Kashiwanoha, Kashiwa, Chiba 277-8564, Japan
[6]Research Institute for Applied Mechanics, Kyushu University
6-1 Kasuga-koen, Kasuga, Fukuoka 816-8580, Japan
[7]Department of Ocean Sciences, Tokyo University of Marine Science and Technology
4-5-7 Konan Minato-ku, Tokyo 108-8477, Japan
[8]Faculty of Environmental Earth Science, Hokkaido University
North 10 West 5 Kita-ku, Sapporo, Hokkaido 060-0810, Japan
*Correspondence to*: Toru Kobari (kobari@fish.kagoshima-u.ac.jp)
**Abstract.** The Kuroshio Current has been thought to be biologically unproductive because of its oligotrophic conditions
and low plankton standing stocks. Even though vulnerable life stages of major foraging fishes risk being entrapped by
frontal eddies and meanders and encountering low food availability, they have life-cycle strategies that include growing
and recruiting around the Kuroshio Current. Here we report that phytoplankton growth and consumption by
microzooplankton is stimulated by turbulent nitrate flux amplified by the Kuroshio Current. Oceanographic
observations demonstrate that the Kuroshio Current topographically enhances significant turbulent mixing and nitrate
influx to the euphotic zone. Graduated nutrient enrichment experiments show that growth rates of phytoplankton and
micro-heterotroph communities were stimulated within the range of the turbulent nitrate flux. Results of dilution
experiments imply significant microzooplankton grazing on phytoplankton. We propose that these rapid and systematic
trophodynamics enhance biological productivity in the Kuroshio.

## 1 Introduction

The Kuroshio Current is the western boundary current of the North Pacific Subtropical Gyre (Qiu, 2001; Hu et al., 2015). The Kuroshio enters the East China Sea from the east of Taiwan and flows along the continental slope until it passes through the Tokara Strait into the western North Pacific (Fig. 1a). The Kuroshio has been thought to be biologically unproductive because ambient nutrient concentrations and plankton standing stocks in its waters are low (Guo, 1991; Hirota, 1995). In spite of such seemingly unproductive conditions, the Kuroshio in the East China Sea (ECS-Kuroshio) is adjacent to major spawning and nursery grounds of foraging species such as sardines (Watanabe et al., 1996), jack mackerel (Sassa et al., 2008), chub mackerel (Sassa and Tsukamoto, 2010), and common squid (Bower et al., 1999). Indeed, good fishing grounds have been found for various fishes and squid near the Kuroshio, and the catches from those grounds account for more than half of the total catch in Japanese waters (Saito, 2019). It is risky, however, for highly vulnerable early life stages of many foraging species to grow and recruit in the oligotrophic and unproductive waters of the ECS-Kuroshio (hereafter called the "Kuroshio Paradox": Saito, 2019), even if the warm temperatures of the Kuroshio Current can enhance cellular metabolic processes and thereby stimulate growth. Conventional wisdom is that survival of these early stages is supported by the high plankton productivity on the continental shelf and in the Kuroshio front (Nakata et al., 1995). However, these areas of high productivity are limited in extent and spatiotemporally highly variable because the Kuroshio Current often meanders (Nakata and Hidaka, 2003). Coastal water masses are sometimes entrapped and transported into the Kuroshio and to more pelagic sites (Nakamura et al., 2006; Kobari et al., 2019). Use of waters in the vicinity of the oligotrophic Kuroshio as a nursery and feeding ground would therefore appear to be a risky strategy unless there is a mechanism that enhances biological production in

the Kuroshio.
There is increasing information about the community structure of phytoplankton and zooplankton in the Kuroshio.
Phytoplankton standing stocks in the Kuroshio consist mainly of picoplankton and nanoplankton, and the predominant
components are haptophytes, diatoms, and unicellular cyanobacteria like *Prochlorococcus* and *Synechococcus*
(Hasegawa et al., 2019; Endo and Suzuki, 2019). Heterotrophic bacteria and calanoid copepods contribute to
heterotrophic biomass in the Kuroshio, whereas microzooplankton biomass is relatively small (Kobari et al., 2019).
Based on a mass balance model, Kobari et al. (2019) have concluded that mesozooplankton standing stocks in the
Kuroshio are supported by micro- and nano-autotrophs and microzooplankton. However, we have little understanding
of how biogeochemical processes and trophodynamics support the plankton community in the Kuroshio.
In recent years, some mechanisms that supply nutrients to the oligotrophic waters of the Kuroshio have been
identified. The Kuroshio "nutrient stream" characterized by an intense core of nutrient flux at subsurface contributes
substantially to productivity in the euphotic zone of the Kuroshio in a manner similar to the contribution of the "nutrient
stream" along the Gulf Stream (Komatsu and Hiroe, 2019). Turbulence around the Kuroshio appears to be an important
mechanism that supplies nutrients via upward movement of deeper waters into the Kuroshio (Nagai et al., 2019).
Frontal disturbances also contribute to the supply of nutrients into the euphotic zone of the Kuroshio (Kuroda, 2019).
Moreover, the Island Mass Effect produced by the Kuroshio Current as it flows over the bottom topography of the
Japanese archipelago induces an upward supply of nutrients (Hasegawa, 2019). These nutrient supplies have been
hypothesized to stimulate biological productivity in the Kuroshio. Within the wide path of the Kuroshio, the supply of
nutrients by these mechanisms can be particularly efficacious around the Tokara Straits because of the extensive frontal
disturbances (Nakamura et al., 2006) and strong turbulence (Tsutsumi et al., 2017; Nagai et al., 2017, 2019) in that area.
Here we report evidence of phytoplankton productivity and subsequent microzooplankton grazing stimulated by
turbulence-induced nitrate fluxes in the Kuroshio Current. Oceanographic observations demonstrated a substantial
nitrate flux caused by turbulent mixing in the Tokara Strait of the ECS-Kuroshio. Nutrient-amended bottle incubation
experiments showed that the growth rates of phytoplankton and micro-heterotrophs, as well as the grazing rates of
microzooplankton on phytoplankton, were elevated within the area impacted by this turbulence-induced nitrate flux.
**2 Materials and methods**
**2.1 Onboard observations and experiments**
All oceanographic observations and bottle incubations were done in the Kuroshio Current where it passes through the
Tokara Strait. Samplings for nitrate concentrations and measurements of turbulent diffusivity were conducted at 14
stations along two transects across the Kuroshio Current (Fig. 1a) during cruises of the T/S *Kagoshima-maru* in
November 2015.
The nitrate profiles were measured with a nitrate sensor (Deep SUNA V2) attached to a SBE 9plus
conductivity-temperature-depth (CTD) system (Sea Bird Electronics). Turbulent diffusivity was estimated from
microstructure measurements made with a microstructure profiler (TurboMAP-L, JFE Advantech Co., Ltd.) and the
equations of Osborn (1980). The profiler was deployed immediately after each CTD cast at the same station. The nitrate
sensor was calibrated with measured nitrate concentrations (Fig. S1). Because the precision of the nitrate sensor in this
study was 0.37 mmol m$^{-3}$ (estimated by Hasegawa et al., 2019), if we had calculated the vertical nitrate gradient from
the raw data, the noise level would have been too high to resolve the normal background nitrate stratification of $O$ ($10^{-1}$
mmol m$^{-4}$). We therefore needed to average the sensor data vertically to reduce the level of noise. The bin-averaged
vertical gradient of the sensor data can be written as follows:
$$\frac{\partial \overline{Cs}}{\partial z} \sim \frac{\overline{Cr}_i - \overline{Cr}_{i-1}}{\Delta z} \pm P\sqrt{\frac{2\overline{w}}{\Delta z^3 f}}$$    (1)
where $Cs$ is the nitrate concentration reported by the sensor, $Cr$ is the real concentration, $\overline{w}$ is the average vertical
deployment speed of the sensor, $f$ is the sampling frequency, and $\Delta z$ is the average bin size. In this study $f = 1$ Hz and $\overline{w}$
$= 0.5$ m s$^{-1}$. The second term on the right side of Eq. (1) indicates the expected precision of the bin-averaged vertical
gradient of nitrate (see the detailed discussions in Hasegawa et al., 2019). In this study, we set $\Delta z = 10$ m to resolve the
likely vertical gradient with an expected imprecision of $O$ ($10^{-2}$ mmol m$^{-4}$).

A total of sixteen nitrate and turbulent diffusivity profiles were averaged at the stations that were studied during the

KG1515 cruise of the T/S *Kagoshima-maru* across the Kuroshio path. The profiles of the gradients of the averaged
nitrate concentrations and averaged turbulent diffusivity were then multiplied at each depth to calculate the average
turbulent nitrate fluxes. Both parameters were binned and averaged within 10-meter intervals. The vertical gradient of
the averaged nitrate profile ($C_{NO3}$) and the averaged vertical diffusivity ($K_z$) were then multiplied at each depth ($z$) to
estimate the area-averaged vertical turbulent nitrate flux ($F_{NO3}$) as follows:
$$F_{NO3} = -K_Z \times \partial C_{NO3} / \partial z$$    (2)

In recent years, there has been a lively discussion about the importance of the diapycnal advective flux associated

with the diffusive flux (e.g., Du et al., 2017). However, in the present study, we assumed that the important nutrient flux
was the flux across the base of the euphotic zone, not the flux through the pycnocline, which can be broken down by
turbulent mixing. In addition, because our study area included frontal regions, unlike the South China Sea where the
Kuroshio flows over seamounts, density fluctuations could have been caused not only by turbulent mixing but also by
advection and the movement of fronts. Accordingly, we focused our analysis on the vertical turbulent nutrient flux using
Cartesian coordinates rather than on the diapycnal flux using isopycnal coordinates.
We performed two different types of bottle incubations. For phytoplankton and micro-heterotrophs, growth rates in
response to in situ nitrate fluxes induced by turbulent mixing were estimated using bottle incubations with nutrient
gradients ($EXP_a$) at eight stations in both November 2016 and November 2017. To estimate microzooplankton grazing
rates on phytoplankton, dilution experiments ($EXP_b$) following the methodology of Landry and Hasset (1982) were
performed at eight stations in November 2017 (Fig. 1b, Table 1).
**2.2 Experimental setup**
Seawater samples for all experiments were obtained using 2.5-L Niskin-X bottles attached to a CTD profiler and
carousel multisampling system (CTD-CMS: SBE 9plus, Sea Bird Electronics). The samples were transferred by gravity
filtration using a silicon tube with a nylon filter (0.1-mm mesh opening) into the incubation bottles for $EXP_a$ and $EXP_b$.
The $EXP_a$ experiment was performed using duplicate 2.3-L polycarbonate bottles without added nutrients and with
a mixture of nitrate ($NaNO_3$) and phosphate ($KH_2PO_4$) in an atomic N:P ratio of 15:1. The nitrate concentrations were
either 0 (control), 0.05, 0.15, 0.5, 0.75, 1.5, or 5 $\mu mol\ L^{-1}$. If the turbulent nitrate influx at the subsurface chlorophyll
maximum observed in the Tokara Strait ($O$: 0.788 mmol $m^{-2}$ $d^{-1}$, see Results) were continued for 5.3 days while the
Kuroshio Current (0.33 m $s^{-1}$, Zhu et al., 2017) passed through the Tokara Strait (150 km), the phytoplankton in a layer
10 m thick could have consumed nitrate equivalent to a nitrate enrichment of 0.41 $\mu mol\ L^{-1}$.
The $EXP_b$ experiment was conducted using triplicate 1.2-L polycarbonate bottles with microzooplankton as
grazers and involved dilutions of the microzooplankton standing stocks in the original water samples so that the
concentrations of microzooplankton equaled 1, 0.6, 0.3, or 0.1 times the concentration in the undiluted water. These
treatment bottles were enriched with 3 µmol $L^{-1}$ nitrate (NaNO$_3$) and 0.2 µmol $L^{-1}$ phosphate (KH$_2$PO$_4$) to promote
phytoplankton growth. In addition, to evaluate nutrient limitation of phytoplankton growth, extra triplicate undiluted
bottles were incubated without nutrient amendments.
All incubation bottles were soaked in 10% HCl and rinsed with surface seawater at each station before use (Landry
et al., 1995). All experimental bottles were incubated for 72 h for EXP$_a$ and 24 h for EXP$_b$ in a water bath with running
surface seawater for temperature control and were covered by nylon mesh screening (i.e., screening with 5-mm
openings) to reduce irradiance to 75% of the surface irradiance. Phytoplankton growth in the incubation bottles might
have been an overestimate of in situ growth because subsurface irradiance was lower than the irradiance in the
incubation bottles.
**2.3 Sample analysis**
Chlorophyll $a$ concentrations were determined at the beginning and end of the EXP$_a$ and EXP$_b$ incubations. Subsamples
of 500–1000 mL were filtered through a nylon mesh (11-µm mesh opening: Millipore NY1104700) and a glass-fiber
filter (2-µm: Whatman GM/F; 0.7-µm: Whatman GF/F) for EXP$_a$ and through a glass-fiber filter (GF/F) for EXP$_b$ at a
pressure less than 20 kPa. Photosynthetic pigments were extracted overnight in $N,N$-dimethylformamide at –20 °C in
the dark, and the chlorophyll $a$ concentrations were determined with a fluorometer (Turner Designs 10AU or TD700).
Size fractions were defined as Pico for chlorophyll in phytoplankton smaller than 2 µm, Nano for chlorophyll in
phytoplankton between 2 and 11 µm in size, and Micro for chlorophyll in phytoplankton larger than 11 µm.
Micro-sized heterotrophs in the incubation bottles at the beginning of $EXP_a$ and $EXP_b$ were examined. Subsamples
of 500 mL were collected and fixed with 3% acid Lugol's solution. We identified and counted three taxonomic groups
of the micro-heterotroph community (naked ciliates, tintinnids and copepod nauplii) with an inverted microscope (Leica
Leitz DMRD). Some marine planktonic ciliates and flagellates are known to be mixotrophs (Gaines and Elbrächter,
1987), but we assumed naked ciliates and tintinnids to be heterotrophic in the present study. The sizes of cells or of
individuals were measured, the biovolume was computed based on geometric shape, and the carbon content was
estimated using conversion equations (Put and Stoecker, 1989; Verity and Langdon, 1984; Parsons et al., 1984).
**2.4 Rate calculations**
Apparent growth rates ($g$: $d^{-1}$) in the incubation bottles of $EXP_a$ and $EXP_b$ were calculated from size-fractionated
chlorophyll $a$ concentrations ($\mu g\ L^{-1}$) or standing stocks ($\mu g\ C\ L^{-1}$) of micro-heterotroph groups identified at the
beginning ($C_o$) and end ($C_t$) of the incubations period ($t$: days):
$g = [\ln(C_t) - \ln(C_o)]/t$                     (3)
$C_t$ in the incubation bottles of $EXP_b$ can be calculated using the following equation (Landry et al., 1995):
$C_t = C_o \times \exp[(g_{max} - m) \times t]$               (4)
where $g_{max}$ and $m$ are the maximum growth rate of size-fractionated phytoplankton ($d^{-1}$) and their mortality rate by
microzooplankton grazing ($d^{-1}$), respectively. The maximum growth rate ($g_{max}$) and mortality rate were determined with
a linear regression of the apparent growth rate ($g$) against dilution factor (X):
$g = g_{max} - mX$                                       (5)
where $m$ is the mortality rate in the undiluted water (X = 1). All parameters derived from $EXP_a$ and $EXP_b$ are listed in
Table 2 and Table 3.
**2.5 Data analysis**
To quantify the sensitivity of phytoplankton growth rates to nutrient supply rates, we calculated the slopes of linear
regressions of growth rates for the size-fractionated chlorophyll $a$ concentrations versus the logarithms of the enriched
nitrate concentrations. We then computed the Pearson correlation coefficient of these slopes to nitrate + nitrite and
phosphate concentrations and microzooplankton biomass at the beginning of each incubation. A one-way analysis of
variance (ANOVA) with a post-hoc Tukey honestly significant difference test was used to compare maximum growth
rates, mortality rates, and net growth rates among the three size fractions.
**3 Results**
**3.1 Oceanographic observations**
Turbulent diffusivity and nitrate concentrations were measured in order to estimate the vertical turbulent nitrate flux
along the transects across the Kuroshio Current in the Tokara Strait, where a shallow ridge lies in the path of the
Kuroshio. We obtained 16 pairs of vertical profiles of turbulent diffusivity and nitrate concentrations and estimated the
averages and 95% confidence intervals of the vertical profiles. The averaged chlorophyll-$a$ profile (Fig. 2a), which was
recorded with a light-emitting diode fluorometer on a TurboMAP-L profiler, revealed a subsurface chlorophyll
maximum (SCM) at 60 m, which was almost coincident with a sharp increase in the nitrate concentration (i.e., the top
of the nitracline). Vertical diffusivity of $O$ ($10^{-4}$ m$^2$ s$^{-1}$, Fig. 2b) was higher at 70 m than at depths of 80–130 m. Just
below the SCM peak, relatively high nitrate concentrations and vertical diffusivity induced vertical turbulent nitrate
fluxes of $O$ (1 mmol m$^{-2}$ d$^{-1}$, Fig. 2c).
**3.2 Gradient enrichment experiments (EXP$_a$)**
To evaluate how the turbulent nitrate fluxes measured in the Tokara Strait increased the standing stocks of
phytoplankton and micro-heterotrophs in the Kuroshio, we conducted bottle incubations of the phytoplankton and
micro-heterotroph communities enriched with different nutrient concentrations (EXP$_a$). The total chlorophyll $a$
concentrations at the beginning of EXP$_a$ averaged among the duplicate samples ranged from 0.15 to 0.52 µg L$^{-1}$ (Table
1). The pico-fractions and nano-fractions accounted for more than 80% of the total chlorophyll $a$ (Fig. 3a). All
size-fractionated chlorophyll $a$ declined or changed little toward the end of the incubations at nitrate enrichments <0.15
µmol L$^{-1}$, but they increased at enrichments >0.5 µmol L$^{-1}$.
At the beginning of the incubations, micro-heterotroph standing stocks averaged among the duplicate samples
ranged from 0.12 to 0.79 µg C L$^{-1}$ (Table 1). Naked ciliates accounted for 51–96% of the micro-heterotrophic biomass
in terms of carbon at the beginning of the incubations. Copepod nauplii were the second greatest contributor to the
micro-heterotroph biomass because of their low abundance but large individual body mass; tintinnid ciliates were a
minor component of the micro-heterotroph biomass. The standing stocks of all taxonomic groups in the
micro-heterotroph category increased with increasing nitrate enrichment (Fig. 3b), but the patterns of increase in
response to nutrient enrichment were less clear than was the case for the size-fractionated chlorophyll $a$ concentrations.
Based on the changes of the standing stocks between the beginning and end of the incubations, we investigated the
growth rates of the chlorophyll and micro-heterotrophs. The growth rates of all size-fractionated chlorophyll increased
at higher concentrations of added nitrate (Fig. 4a). Growth rates were negative or close to zero for all size-fractions at
nitrate enrichments <0.15 µmol L$^{-1}$. However, the growth rates of the pico- and micro-sized chlorophyll were positive
at nitrate enrichments >0.5 µmol L$^{-1}$, which were nearly equivalent to the concentrations associated with the turbulent
nitrate fluxes observed in the Tokara Strait (see section 2.2). Because micro-heterotroph growth rates varied among
stations, the response of micro-heterotroph growth to the nutrient enrichments was ambiguous (Fig. 4b). Growth rates
were positive for copepod nauplii at all nitrate enrichments and were higher for both naked and tintinnid ciliates at
higher nitrate enrichments. Thus, the standing stocks of phytoplankton and micro-heterotrophs were likely increased by
additions of nitrate within the range of fluxes measured in the Tokara Strait.

The slope of a linear regression of the growth rates of the size-fractionated chlorophyll and the logarithms of the

nitrate enrichments for each incubation provided a metric of the sensitivity of phytoplankton growth rates to nutrient
supplies. The steeper slopes at some stations in the upstream Kuroshio in the Tokara Strait compared to the slopes at
other stations (Fig. S2) suggested that the apparent phytoplankton growth rates varied with nutrient concentrations or
predatory impacts at the beginning of the incubations. To determine whether growth rates of the size-fractionated
chlorophyll might have varied with initial nutrient concentrations (bottom-up control) or predator biomasses (top-down
control) at the beginning of the experiments, we compared the slopes to the nitrate + nitrite concentrations (Fig. 5a),
phosphate concentrations (Fig. 5b), and micro-heterotroph biomasses (Fig. 5c) in the ambient seawater without
enrichment. No significant correlation was found between the micro-heterotrophic biomass and the rate of change of
any size-fractionated chlorophyll. In contrast, the fact that there was a negative correlation between the slopes for all
size fractions and the nitrate + nitrite or phosphate concentrations indicated that the stimulation of the phytoplankton
growth rates by nutrients was greater for all chlorophyll size fractions under more oligotrophic conditions. Thus, the
variations of phytoplankton growth rates were likely associated with nutrient concentrations at the beginning of the
incubations.
**3.3 Dilution experiments (EXP$_b$)**
To evaluate how much each size-fractionated phytoplankton population was removed by microzooplankton grazing, we
conducted dilution experiments concurrently with the gradient enrichment experiments. The maximum growth rates
(i.e., the intercepts of the regressions corresponding to $X = 0$ in Eq. (5)) were relatively high for the nano-chlorophyll
(Fig. 6a), but the differences were insignificant among the three size fractions (ANOVA, $p > 0.05$). These results
indicated that the growth potential in the absence of microzooplankton grazing was similar for the nano-sized
chlorophyll compared with the pico- and micro-fractions. In contrast, the slopes of the regressions are the mortality
rates due to microzooplankton grazing, and the fact that they were significantly higher for the nano-chlorophyll versus
the pico- and micro-chlorophyll (ANOVA + Tukey, $p < 0.05$) indicated that the microzooplankton preferentially grazed
on the nano-chlorophyll.

To evaluate the impact of microzooplankton grazing on phytoplankton growth, we compared three different net

growth rates: the observed net growth rates without enrichment ($g_o$), the net growth rates with enrichment ($g_{en}$) in the
undiluted bottles, and the net growth rates ($g_{en}'$) estimated by subtracting the mortality rate ($m$) from the maximum
growth rates ($g_{max}$). For all size-fractionated chlorophyll, the fact that $g_o$ was lower than $g_{en}$ (Fig. 7) indicated that net
growth rates were limited by nutrients. The values of $g_{en}$ and $g_{en}'$ were comparable, i.e., there was no significant
difference between the two (Welch's $t$-test). These results implied that the $g_{en}$ of all size-fractionated chlorophyll could
balance microzooplankton grazing mortality by growing at the maximum rate. In the case of the nano-chlorophyll, the
net growth rates were a bit low because the mortality rates due to microzooplankton grazing exceeded the maximum
growth rates.
**4 Discussion**
The Kuroshio Current impinges on numerous shallow ridges with small islands and seamounts in the Tokara Strait.
Several studies have pointed out that those steep topographic features stir and modify the water column through
upwelling (Hasegawa et al., 2004, 2008) and turbulent mixing (Tsutsumi et al., 2017; Nagai et al., 2017). Compared
with the turbulent nitrate fluxes reported in previous studies, the fluxes observed in the Tokara Strait were one order of
magnitude higher than those reported in the Kuroshio Extension front (Kaneko et al., 2012, 2013; Nagai et al., 2017),
much greater than those at other oceanic sites, and equivalent to those at coastal sites (Cyr et al., 2015). The turbulent
nitrate flux in the downstream Kuroshio Current near the Tokara Strait is similar in magnitude to our estimates (Nagai et
al., 2019). Because the Kuroshio Current runs steadily through the Tokara Strait, this nutrient supply induced by
turbulent diffusivity is considered to be one of the mechanisms that enhance phytoplankton productivity, even under
oligotrophic conditions in the Kuroshio Current.
Despite the large turbulent nitrate flux ($O$: 1 mmol m$^{-2}$ d$^{-1}$), the chlorophyll $a$ concentrations in the area of the
Tokara Strait impacted by the Kuroshio Current were as low as the values reported from nearby areas of the Kuroshio
(Kobari et al., 2018, 2019) and oceanic sites in the North Pacific Ocean (Calbet and Landry, 2004). Based on the
gradient enrichment experiments, standing stocks and the growth rates of all size-fractionated phytoplankton increased
at nitrate enrichments above 0.5 µmol L$^{-1}$, which were equivalent to the concentrations produced by the observed
turbulent nitrate flux. These results suggest that phytoplankton standing stocks and growth rates are stimulated by the
magnitude of the observed turbulent nitrate flux.

In global comparisons, microzooplankton grazing has a significant impact on phytoplankton, particularly at oceanic

sites (Calbet and Landry, 2004). Microzooplankton standing stocks in the Kuroshio Current as it passes through the
Tokara Strait are lower than those on the continental shelf of the ECS (Chen et al., 2003) and might be removed by
mesozooplankton predation (Kobari et al., 2019). The low microzooplankton standing stocks in the Kuroshio Current
imply low microzooplankton grazing on phytoplankton. However, the dilution experiments demonstrated that
phytoplankton mortality by microzooplankton grazing was high and equivalent to 41–122% of the maximum growth
rates of the phytoplankton, based on the ratio of the mortality rate to the maximum growth rates of total chlorophyll *a*
(Table 2). Indeed, phytoplankton could likely balance microzooplankton grazing mortality by growing at maximum
rates, particularly in the case of the nano-phytoplankton (Fig. 7). These results from concurrently conducted
experiments suggested that phytoplankton standing stocks are stimulated by turbulent nitrate fluxes and are then quickly
removed by microzooplankton grazing, particularly in the case of nanophytoplankton. Taking into account the size
range of prey for ciliates (Pierce and Turner, 1992) and copepod nauplii (Uye and Kasahara, 1983), microzooplankton
grazing could be a major reason why phytoplankton do not attain high growth rates and standing stocks, even when
their growth potential is high and they are sensitive to nutrient enrichments. The rapid transfer of the elevated
phytoplankton production to microzooplankton might thus be a possible explanation for the low chlorophyll
concentrations, even when there are large turbulent nitrate fluxes in the Kuroshio Current.

The standing stocks and growth rates of all micro-heterotrophs were relatively high in the higher nitrate

enrichments, but the patterns of increase were less clear than in the case of the phytoplankton. This difference was
probably due to the large variations in the micro-heterotroph standing stocks among stations (Table 1) and slower
growth than phytoplankton. Indeed, the lack of clarity of this pattern was remarkable for the copepod nauplii because of
their relatively slow growth rates, lower abundance in the bottles, and larger individual body masses. In contrast,
"intra-guild" predation within micro-heterotroph communities might be another explanation for the ambiguous pattern
of their standing stocks and growth rates. The growth rates of copepod nauplii were always higher than those of naked
ciliates, especially when there was no or little nitrate supplied. The ratio of mean equivalent spherical diameter of body
mass for copepod nauplii (88 μm) and naked ciliates (16 μm) was estimated to be 5:1, which is much different from a
typical predator–prey mass ratio (i.e., 18:1, Hansen et al., 1994). Thus, it is unlikely that intraguild predation of copepod
nauplii on naked ciliates would happen in the bottles. More importantly to the ambiguous pattern of the growth of the
micro-heterotrophs, the results from the concurrently conducted experiments implied that phytoplankton productivity
was stimulated by the turbulent nitrate flux and the phytoplankton rapidly grazed by microzooplankton, but standing
stocks and growth rates of micro-heterotrophs were not elevated during three days in the Kuroshio Current. An increase
of micro-heterotroph standing stocks and their trophic transfer to mesozooplankton might have been apparent further
downstream in the Kuroshio Current.

There is increasing evidence that turbulence-induced nutrient fluxes promote phytoplankton growth in the open

ocean (Kaneko et al., 2013; Nagai et al., 2017, 2019). However, there is no experimental documentation for a response
of the phytoplankton community to this nutrient supply or of subsequent trophic transfer in a planktonic food web. In
the tropical and subtropical oceans, microzooplankton grazing has been thought to be a major source of phytoplankton
mortality and has been shown to account for more than 75% of phytoplankton daily growth (Calbet and Landry, 2004).
Furthermore, strong trophic linkages are well known between microbes and metazoans through microzooplankton
(Calbet and Landry, 1999; Calbet et al., 2001; Calbet and Saiz, 2005; Kobari et al., 2010). Our study has provided the
first experimental evidence that phytoplankton standing stocks and growth rates are stimulated by turbulent nutrient
fluxes and rapidly grazed by microzooplankton. These results imply that biological productivity may be underestimated
because of the apparently low nutrient concentrations and low phytoplankton biomass in the Kuroshio. Because of the
strong turbulence amplified by the Kuroshio Current, phytoplankton productivity stimulated by nutrient fluxes and rapid
trophic transfer to microzooplankton are likely to happen in the Tokara Strait and downstream. We therefore propose
that undocumented biological productivity in the Kuroshio is sustained by these rapid and systematic trophodynamics.
Such undocumented biological production, elevated by the rapid and systematic trophodynamics, may provide a good
supply of food for the vulnerable stages of foraging fishes around the Kuroshio and thus explain part of the Kuroshio
Paradox.

**Data Availability Statement:**
All relevant data are shown in the paper as tables and figure.

**Author Contributions**
T. Kobari, DH and NY conceived and designed the oceanographic observations and experiments. DH, HN, AN,
ET, TM, TN performed the oceanographic observations and turbulence measurements. T. Kobari, TH, T. Kanayama and
FK performed the onboard experiments. T. Kobari, TH, T. Kanayama, FK, NY, KS analyzed the samples and data of the
onboard experiments. DH and TT analyzed the data of oceanographic observations and turbulence measurements. T.
Kobari, GK, HN and XG organized the research cruises.

**Competing interests:**
The authors declare no competing or conflict interests.

**Acknowledgements**
We thank the captains and crew of the T/S *Kagoshima-maru* for their help in oceanographic observations and
sample collections.

**Financial support:**
This study has been supported by grants from the Japan Society for the Promotion of Science (17K00522,
18H04920, 4702), Ministry of Education, Culture, Sports, Science and Technology in Japan (The Study of Kuroshio
Ecosystem Dynamics for Sustainable Fisheries).

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

**Table 1** Location and environmental conditions at the stations in the ECS-Kuroshio where gradient enrichment ($EXP_a$)
and dilution experiments ($EXP_b$) were conducted. Depth: sampling depth (m) of water samples for each experiment.
WT: mean water temperature during the experiments (ºC). $NUTs_o$: nutrients concentrations ($\mu$mol $L^{-1}$) at the beginning
of each experiment. $CHL_o$: Chlorophyll $a$ concentration ($\mu$gCHL $L^{-1}$) at the beginning of the experiments. $MiZ_o$:
micro-heterotroph standing stock at the beginning of each experiment ($\mu$gC $L^{-1}$). DL: below the detection limit.

| Station | Location | | Date | Year | Depth | WT | $NUTs_o$ | | $CHL_o$ | $MiZ_o$ |
|---|---|---|---|---|---|---|---|---|---|---|
| | Longitude | Latitude | | | | | $NO_3+NO_2$ | $PO_4$ | | |
| $EXP_a$ | | | | | | | | | | |
| C02 | 30°11'N | 129°41.0'E | 13 Nov | 2016 | 68 | 26.1 | DL | 0.02 | 0.34 | 0.19 |
| C03 | 29°50'N | 129°08.4'E | 13 Nov | 2016 | 75 | 26.2 | DL | 0.01 | 0.41 | 0.27 |
| F01 | 29°53'N | 129°22.4'E | 14 Nov | 2016 | 81 | 25.1 | 0.21 | 0.04 | 0.35 | 0.15 |
| G01 | 29°51'N | 129°57.2'E | 14 Nov | 2016 | 91 | 26.1 | 0.26 | 0.07 | 0.44 | 0.12 |
| K02 | 29°34'N | 128°26.3'E | 12 Nov | 2017 | 50 | 25.6 | 0.18 | DL | 0.31 | 0.23 |
| K05 | 30°06'N | 130°11.9'E | 14 Nov | 2017 | 105 | 24.8 | 0.57 | 0.02 | 0.52 | 0.79 |
| K08 | 30°24'N | 131°23.6'E | 15 Nov | 2017 | 115 | 25.5 | 1.82 | 0.12 | 0.15 | 0.34 |
| K11 | 31°24'N | 132°29.2'E | 16 Nov | 2017 | 90 | 25.0 | 0.16 | DL | 0.27 | 0.55 |
| $EXP_b$ | | | | | | | | | | |
| A05a | 30°10'N | 129°17.5'E | 3 Nov | 2017 | 13 | 25.5 | 0.10 | 0.03 | 0.23 | 0.12 |
| A05b | 30°10'N | 129°17.5'E | 7 Nov | 2017 | 95 | 25.5 | DL | DL | 0.16 | 0.15 |
| A05c | 30°11'N | 129°17.2'E | 7 Nov | 2017 | 34 | 25.3 | 0.02 | 0.01 | 0.24 | 0.05 |
| A06a | 30°00'N | 129°15.1'E | 3 Nov | 2017 | 12 | 25.4 | DL | 0.02 | 0.16 | 0.13 |
| A06b | 30°00'N | 129°15.0'E | 7 Nov | 2017 | 110 | 25.7 | 1.61 | 0.11 | 0.14 | 0.04 |
| A08a | 29°19'N | 129°09.4'E | 6 Nov | 2017 | 76 | 25.6 | DL | 0.02 | 0.29 | 0.22 |
| A08b | 29°26'N | 129°12.4'E | 6 Nov | 2017 | 71 | 25.6 | 0.03 | 0.01 | 0.21 | 0.17 |
| A09a | 29°09'N | 129°00.0'E | 6 Nov | 2017 | 105 | 25.6 | 0.11 | 0.02 | 0.20 | 0.15 |


**Table 2** Phytoplankton growth rate ($d^{-1}$) derived from the gradient enrichment experiments ($EXP_a$) in the ECS-Kuroshio.
Enriched nitrate concentrations ($\mu mol\ L^{-1}$) are shown at the top of each column. A and B: duplicate bottles. Pico:
chlorophyll smaller than 2 μm. Nano: chlorophyll between 2 and 11 μm. Micro: chlorophyll larger than 11 μm.

| Station | 0 A | 0 B | 0.05 A | 0.05 B | 0.15 A | 0.15 B | 0.5 A | 0.5 B | 0.75 A | 0.75 B | 1.5 A | 1.5 B | 5 A | 5 B |
|---|---|---|---|---|---|---|---|---|---|---|---|---|---|---|
| **Micro** | | | | | | | | | | | | | | |
| C02 | -0.108 | -0.116 | -0.089 | -0.082 | 0.019 | -0.073 | 0.470 | 0.426 | 0.422 | 0.441 | 0.686 | 0.798 | 0.796 | 0.556 |
| C03 | -0.116 | -0.118 | -0.073 | -0.078 | -0.004 | -0.008 | 0.453 | 0.426 | 0.588 | 0.706 | 0.780 | 0.892 | 0.862 | 0.906 |
| F01 | 0.150 | 0.159 | 0.332 | 0.277 | 0.282 | 0.344 | 0.445 | 0.495 | 0.511 | 0.497 | 0.490 | 0.385 | 0.372 | 0.467 |
| G01 | 0.062 | 0.051 | 0.135 | 0.089 | 0.163 | 0.108 | 0.438 | 0.477 | 0.795 | 0.736 | 0.828 | 0.969 | 0.861 | 0.781 |
| K02 | -0.305 | -0.282 | -0.205 | -0.265 | -0.113 | -0.305 | 0.264 | 0.295 | 0.119 | 0.097 | 0.422 | 0.652 | 0.831 | 0.669 |
| K05 | -0.147 | 0.027 | 0.007 | -0.053 | 0.037 | 0.084 | 0.329 | 0.176 | 0.263 | 0.168 | 0.645 | 0.716 | 0.792 | 0.701 |
| K08 | 0.348 | 0.266 | 0.350 | 0.315 | 0.333 | 0.407 | 0.361 | 0.185 | 0.448 | 0.416 | 0.377 | 0.468 | 0.403 | 0.417 |
| K11 | -0.062 | -0.036 | -0.105 | -0.092 | 0.043 | -0.081 | 0.193 | 0.179 | 0.514 | 0.390 | 0.765 | 0.730 | 0.469 | 0.558 |
| **Nano** | | | | | | | | | | | | | | |
| C02 | -0.479 | -0.260 | -0.208 | -0.409 | -0.297 | -0.345 | -0.050 | 0.144 | 0.173 | 0.151 | 0.249 | 0.333 | 0.330 | 0.264 |
| C03 | -0.275 | -0.261 | -0.211 | -0.257 | -0.080 | -0.206 | 0.113 | 0.031 | 0.247 | 0.192 | 0.363 | 0.355 | 0.288 | 0.256 |
| F01 | -0.244 | -0.154 | -0.286 | -0.092 | -0.025 | 0.101 | 0.182 | 0.050 | 0.148 | 0.039 | 0.015 | 0.056 | 0.104 | 0.105 |
| G01 | -0.304 | -0.172 | -0.313 | -0.189 | -0.165 | -0.117 | -0.063 | -0.178 | 0.100 | 0.001 | 0.286 | 0.325 | 0.369 | 0.053 |
| K02 | -0.321 | -0.149 | -0.384 | -0.152 | 0.022 | 0.035 | 0.223 | 0.251 | -0.027 | -0.135 | 0.433 | 0.229 | 0.559 | 0.523 |
| K05 | -0.389 | -0.318 | -0.680 | -0.546 | -0.267 | -0.394 | -0.484 | -0.248 | -0.407 | -0.458 | 0.053 | -0.034 | 0.102 | 0.196 |
| K08 | 0.353 | 0.244 | 0.508 | 0.472 | 0.455 | 0.436 | 0.406 | 0.397 | 0.473 | 0.369 | 0.408 | 0.546 | 0.380 | 0.384 |
| K11 | -0.138 | -0.088 | -0.257 | -0.243 | -0.134 | -0.293 | 0.073 | 0.026 | 0.175 | 0.201 | 0.296 | 0.312 | 0.434 | 0.501 |
| **Pico** | | | | | | | | | | | | | | |
| C02 | -0.383 | -0.188 | -0.186 | -0.199 | -0.119 | -0.162 | 0.188 | 0.143 | 0.162 | 0.241 | 0.257 | 0.291 | 0.377 | 0.205 |
| C03 | -0.202 | -0.258 | -0.259 | -0.282 | -0.143 | -0.160 | 0.017 | -0.019 | 0.148 | 0.191 | 0.194 | 0.248 | 0.230 | 0.300 |
| F01 | -0.071 | -0.091 | -0.054 | -0.032 | 0.050 | 0.129 | 0.205 | 0.144 | 0.216 | 0.141 | 0.170 | 0.134 | 0.031 | 0.172 |
| G01 | 0.019 | -0.061 | 0.051 | -0.032 | 0.019 | 0.008 | 0.156 | 0.162 | 0.323 | 0.188 | 0.338 | 0.308 | 0.344 | 0.366 |
| K02 | -0.245 | -0.253 | -0.257 | -0.275 | -0.243 | -0.230 | -0.046 | 0.010 | -0.067 | -0.101 | 0.065 | -0.030 | 0.203 | 0.089 |
| K05 | -0.087 | 0.031 | 0.014 | -0.027 | 0.103 | 0.157 | 0.057 | 0.261 | 0.130 | 0.339 | 0.316 | 0.255 | 0.368 | 0.404 |
| K08 | 0.032 | 0.055 | -0.013 | 0.228 | 0.262 | 0.201 | 0.240 | 0.069 | 0.262 | 0.281 | 0.177 | 0.284 | 0.222 | 0.327 |
| K11 | -0.197 | -0.216 | -0.194 | -0.146 | -0.046 | -0.071 | -0.005 | 0.033 | 0.163 | 0.076 | 0.236 | 0.049 | 0.092 | 0.179 |


Table 3 Parameters derived from the dilution experiments (EXP$_b$) in the ECS-Kuroshio. $g_{max}$: maximum growth rate ($d^{-1}$). m: mortality rate by microzooplankton grazing ($d^{-1}$). $g_o$: net growth rate measured in the non-enriched and non-diluted bottles ($d^{-1}$). $g_{en}$: net growth rate measured in the enriched and non-diluted bottles ($d^{-1}$). $r^2$: coefficient of determination defined from the linear regression of the apparent growth rate of total chlorophyll $a$ concentrations against dilution factors. p: p-value. Pico: chlorophyll smaller than 2 μm. Nano: chlorophyll between 2 and 11 μm. Micro: chlorophyll larger than 11 μm. Total: total chlorophyll from pico- to micro.

| Station | Pico | | | | Nano | | | | Micro | | | | Total | | | | | |
|---|---|---|---|---|---|---|---|---|---|---|---|---|---|---|---|---|---|---|
| | $g_{max}$ | m | $g_o$ | $g_{en}$ | $g_{max}$ | m | $g_o$ | $g_{en}$ | $g_{max}$ | m | $g_o$ | $g_{en}$ | $g_{max}$ | m | $g_o$ | $g_{en}$ | $r^2$ | p |
| A05a | 0.283 | 0.887 | 0.415 | 0.681 | 1.181 | 1.345 | -0.267 | 0.181 | 0.913 | 0.962 | 0.059 | 0.045 | 1.059 | 0.619 | 0.199 | 0.492 | 0.757 | <0.01 |
| A05b | 0.931 | 1.106 | -0.109 | 0.279 | 1.354 | 1.050 | -0.505 | -0.239 | 0.477 | 0.583 | -0.030 | 0.107 | 1.073 | 1.051 | -0.232 | 0.113 | 0.901 | <0.01 |
| A05c | 0.501 | 0.647 | -0.025 | 0.190 | 1.298 | 1.192 | -0.183 | -0.066 | 0.313 | 0.500 | -0.269 | 0.201 | 0.828 | 0.752 | -0.074 | 0.122 | 0.875 | <0.01 |
| A06a | 0.179 | 0.814 | 0.440 | 0.646 | 0.865 | 1.270 | 0.247 | 0.341 | 0.232 | 0.597 | -0.315 | 0.339 | 0.941 | 0.381 | 0.347 | 0.550 | 0.541 | <0.01 |
| A06b | 0.648 | -0.398 | -0.869 | -1.020 | 0.947 | 0.247 | -0.789 | -0.629 | -0.118 | -0.037 | -0.038 | 0.065 | -0.052 | 0.711 | -0.735 | -0.714 | 0.750 | <0.01 |
| A08a | 0.434 | 0.458 | -0.097 | 0.035 | 1.448 | 1.289 | -0.072 | -0.150 | 0.401 | 0.564 | -0.537 | 0.181 | 0.765 | 0.775 | -0.113 | 0.009 | 0.856 | <0.01 |
| A08b | 0.370 | 0.846 | -0.040 | 0.509 | 0.652 | 1.068 | -0.259 | 0.430 | 0.553 | 1.122 | -0.620 | 0.529 | 0.937 | 0.471 | -0.123 | 0.488 | 0.693 | <0.01 |
| A09a | 0.488 | 0.417 | -0.399 | -0.026 | 0.894 | 0.734 | -0.182 | -0.082 | 0.353 | 0.022 | -0.474 | -0.235 | 0.526 | 0.640 | -0.324 | -0.052 | 0.760 | <0.01 |

**Table 4** Parameters derived from relationship of phytoplankton growth rates against logarithmically transformed
concentrations of enriched nitrate in the gradient enrichment experiments (EXP$_a$). Slope: sensitivity of phytoplankton
growth rate to logarithmically transformed concentrations of enriched nitrate. Intercept: growth potential at the low
nitrate concentration. $r^2$: coefficient of determination defined from the linear regression of growth rate of
size-fractionated chlorophyll $a$ concentrations against logarithmically transformed concentrations of enriched nitrate.
Pico: chlorophyll smaller than 2 µm. Nano: chlorophyll between 2 and 11 µm. Micro: chlorophyll larger than 11 µm.

| Station | Pico | | | Nano | | | Micro | | |
|---|---|---|---|---|---|---|---|---|---|
| | Slope | Intercept | $r^2$ | Slope | Intercept | $r^2$ | Slope | Intercept | $r^2$ |
| C02 | 0.281 | 0.178 | 0.848 | 0.370 | 0.131 | 0.831 | 0.458 | 0.492 | 0.846 |
| C03 | 0.295 | 0.121 | 0.922 | 0.308 | 0.177 | 0.830 | 0.560 | 0.611 | 0.914 |
| F01 | 0.074 | 0.129 | 0.317 | 0.120 | 0.067 | 0.420 | 0.077 | 0.430 | 0.368 |
| G01 | 0.203 | 0.243 | 0.866 | 0.272 | 0.085 | 0.688 | 0.448 | 0.657 | 0.817 |
| K02 | 0.213 | -0.014 | 0.883 | 0.364 | 0.233 | 0.726 | 0.531 | 0.353 | 0.872 |
| K05 | 0.188 | 0.251 | 0.772 | 0.355 | -0.165 | 0.729 | 0.419 | 0.439 | 0.843 |
| K08 | 0.070 | 0.231 | 0.242 | -0.038 | 0.426 | 0.213 | 0.045 | 0.386 | 0.162 |
| K11 | 0.167 | 0.077 | 0.750 | 0.394 | 0.201 | 0.943 | 0.403 | 0.409 | 0.744 |


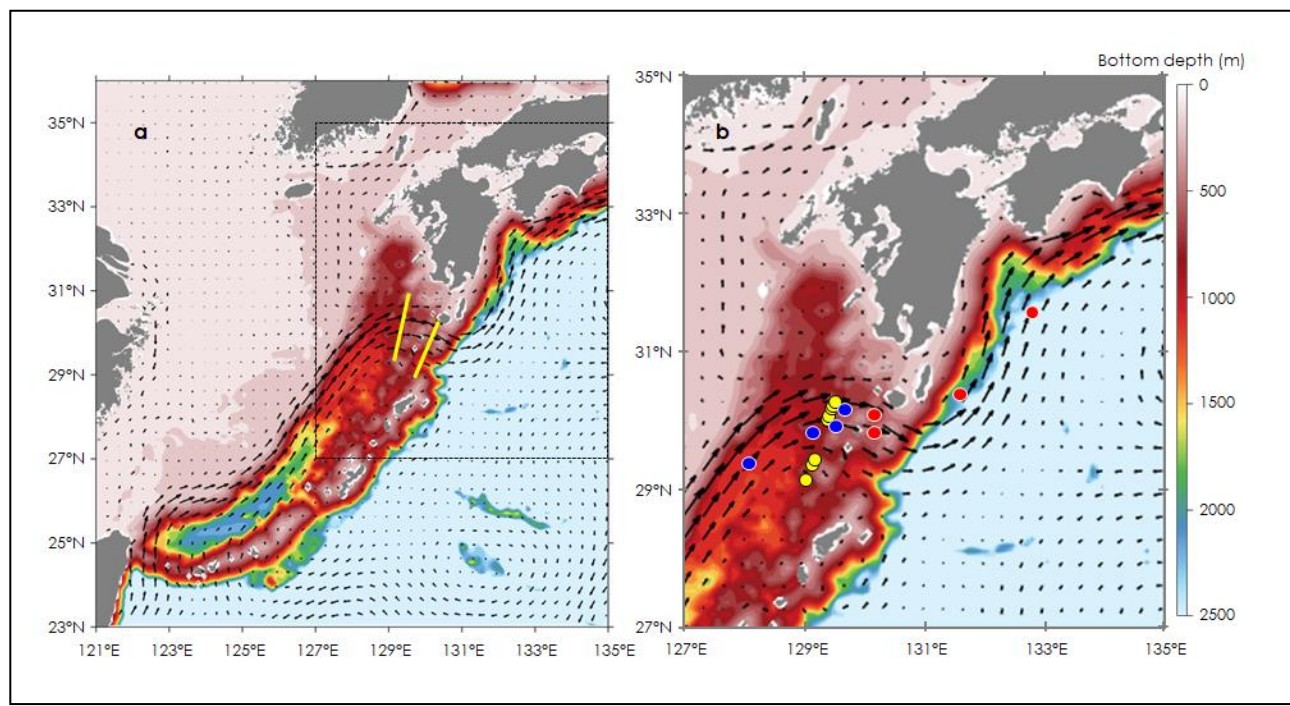

**Figure 1** Locations for oceanographic observations and onboard experiments in the Kuroshio Current of the East China Sea (ECS-Kuroshio). **(a)** Oceanographic observations by Deep SUNA V2 and TurboMAP-L (yellow lines). **(b)** Onboard experiments for phytoplankton and microzooplankton growth (EXP$_a$: red and blue circles) and for microzooplankton grazing (EXP$_b$: yellow circles). EXP$_a$ was conducted in the upstream (blue circles) and downstream Kuroshio (red circles) in the Tokara Strait. Current directions and velocities (arrows) are shown as monthly means during November 2016. Bottom depth (m) is indicated as colored contours.

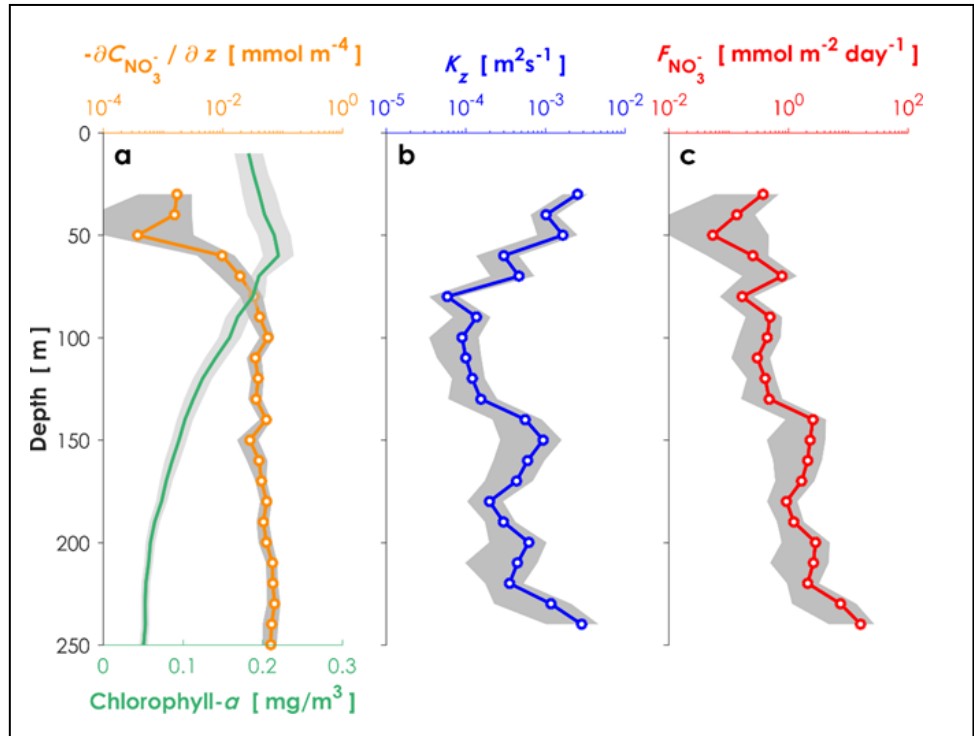


**Figure 2** Vertical profiles of environmental conditions in the Kuroshio Current. **(a)** Nitrate gradient curve (orange) and
chlorophyll *a* concentrations (green) measured with a nitrate sensor (Deep SUNA V2) attached to an SBE-9plus CTD
system. **(b)** Turbulent diffusivity measured with a TurboMAP-L (blue). **(c)** Calculated turbulent nitrate fluxes (red) in
the ECS-Kuroshio. The shaded areas are the 95 percent confidence intervals obtained by a bootstrap process.

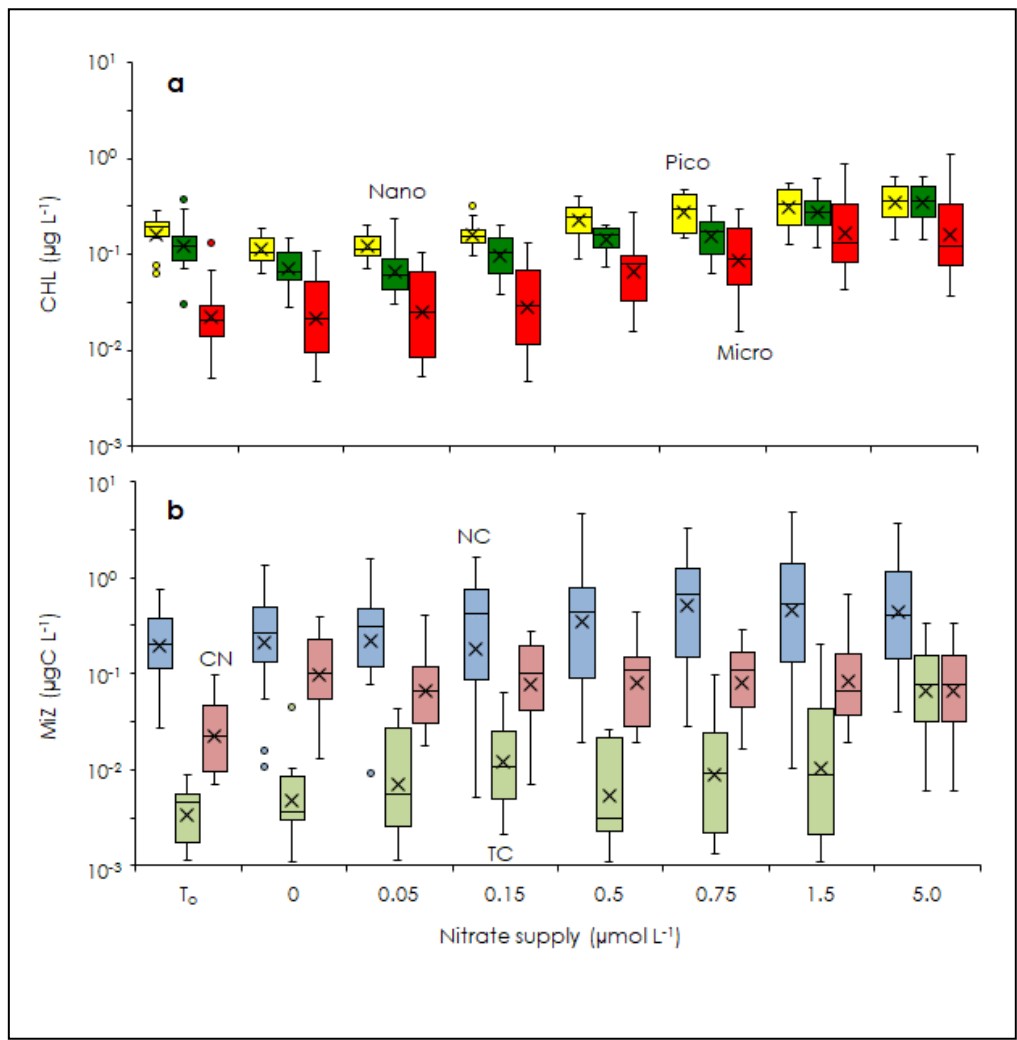


**Figure 3** Changes in phytoplankton and micro-sized heterotroph standing stocks during the gradient enrichment experiments (EXP$_a$). **(a)** Size-fractionated chlorophyll *a* concentrations (CHL). **(b)** Micro-heterotroph standing stocks (MiZ). T$_o$: at the beginning of the gradient enrichment experiments. 0: no enrichment. 0.05 to 5.0 µmol L$^{-1}$: enrichment. Box-and-whisker diagram at each nitrate concentration was compiled from the results conducted at the 8 stations. Box represents first (bottom), second (bar) and third (top) quartiles, and cross marks are the average values. Whiskers indicate minimum and maximum values, and circles are outliers. Pico: chlorophyll smaller than 2 µm (yellow). Nano: chlorophyll between 2 and 11 µm (green). Micro: chlorophyll larger than 11 µm (red). NC: naked ciliates (light blue). TC: tintinnid ciliates (light green). CN: copepod nauplii (light pink).

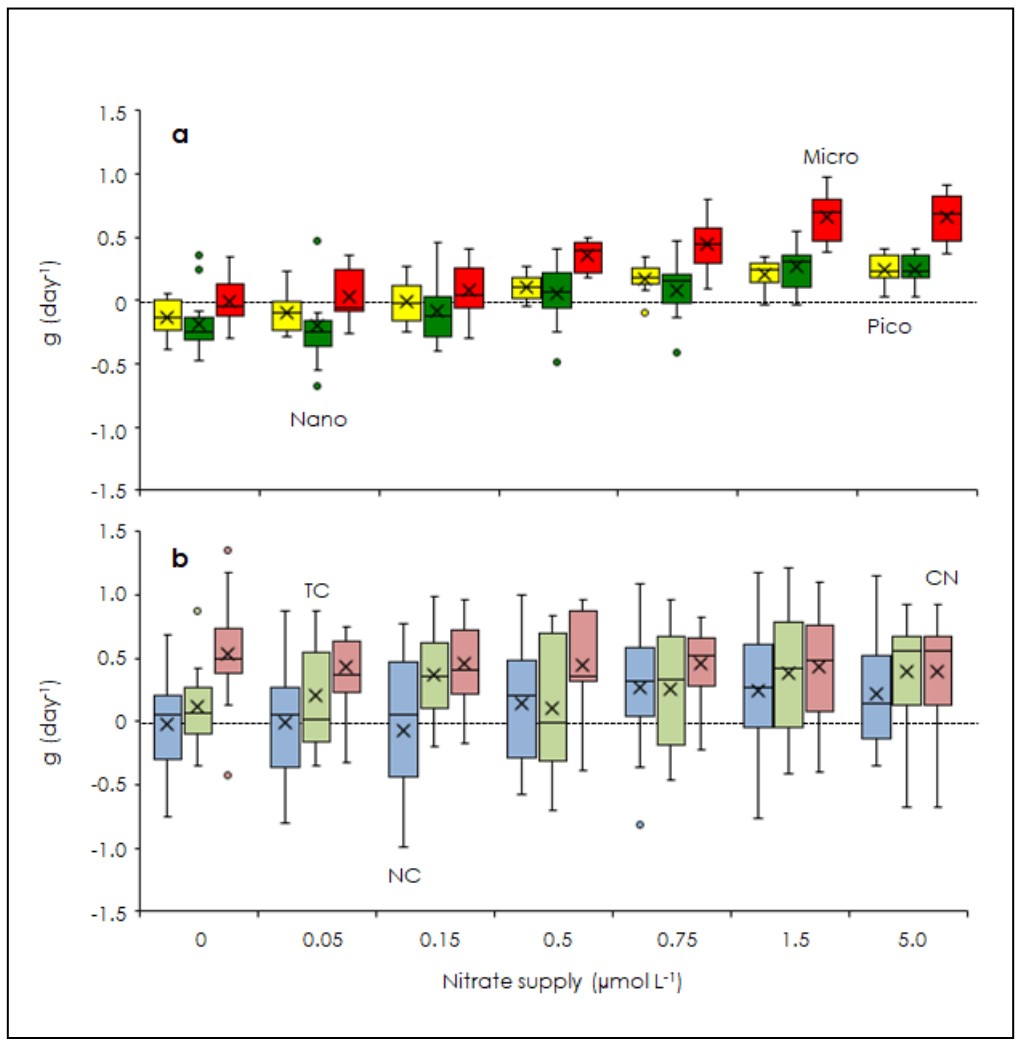

**Figure 4** Changes in phytoplankton and micro-sized heterotroph growth rates in response to nitrate enrichments in the gradient enrichment experiments (EXP$_a$). **(a)** Growth rates (g: d$^{-1}$) of size-fractionated chlorophyll. **(b)** Micro-heterotroph growth rates (g: d$^{-1}$). 0: no enrichment. 0.05 to 5.0 µmol L$^{-1}$: enrichment. Box-and-whisker diagram at each nitrate concentration is based on the results conducted at the eight stations. The symbols have the same meaning as in Figure 3.

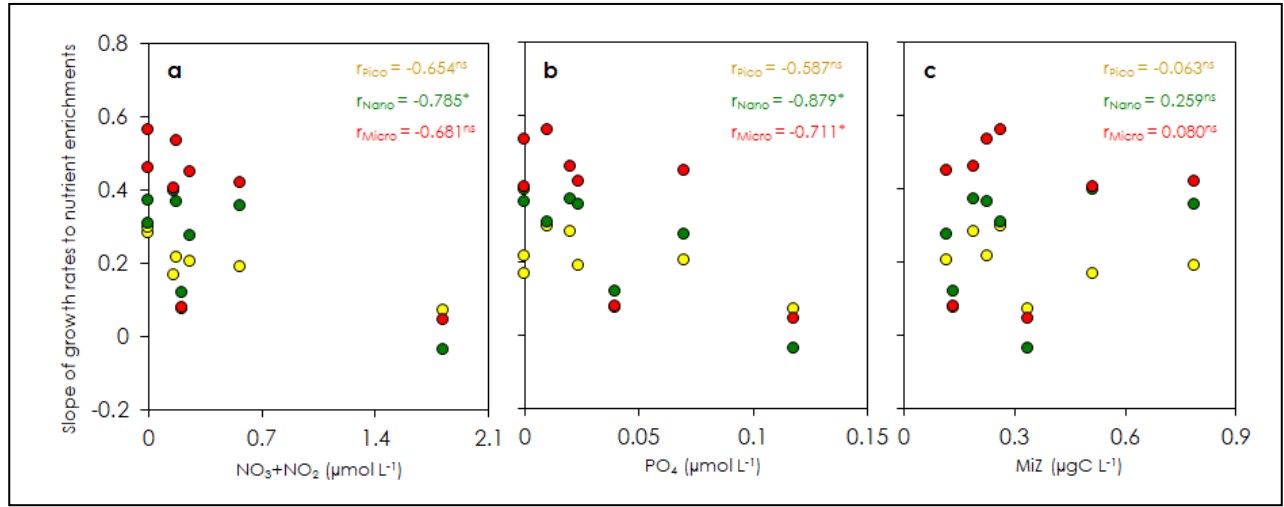

496

**Figure 5** Correlation of the regression slopes of phytoplankton growth rates to nutrient concentrations and micro-sized heterotroph biomass at the beginning of the gradient enrichment experiments ($EXP_a$). **(a)** Regression slopes of the size-fractionated phytoplankton growth versus the concentrations of nitrate ($NO_3$) plus nitrite ($NO_2$). **(b)** Regression slopes of the size-fractionated phytoplankton growth versus the phosphate concentrations ($PO_4$). **(c)** Regression slopes of the size-fractionated phytoplankton growth versus the micro-heterotroph biomass (MiZ). $r$: Pearson correlation coefficient. Pico: chlorophyll smaller than 2 μm. Nano: chlorophyll between 2 and 11 μm. Micro: chlorophyll larger than 11 μm. *: $p < 0.05$. ns: not significant.

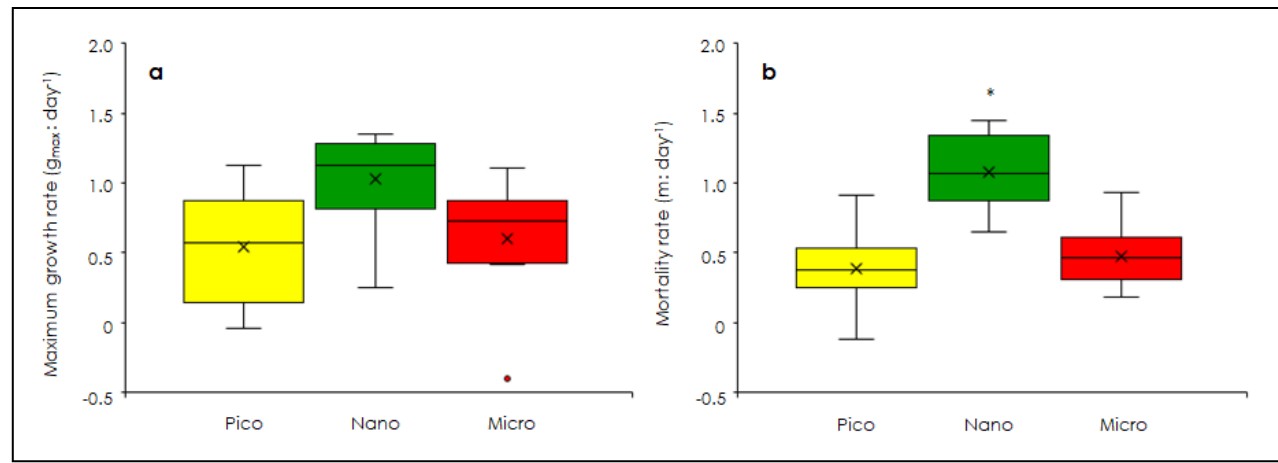


**Figure 6** Comparisons of phytoplankton growth and mortality rates among the three size-fractionated chlorophylls
derived from the dilution experiments (EXP$_b$). **(a)** Maximum growth rates ($g_{max}$). **(b)** Mortality rates by
mirozooplankton grazing. Box-and-whisker diagram at each nitrate concentration was compiled from the results
conducted at the 8 stations. Box represents first (bottom), second (bar) and third (top) quartiles, and cross marks are the
average values. Whiskers indicate minimum and maximum values, and circles are outliers. Asterisk means significant
difference among the three size-fractions (ANOVA + Tukey, $p < 0.05$). Pico: chlorophyll smaller than 2 μm. Nano:
chlorophyll between 2 and 11 μm. Micro: chlorophyll larger than 11 μm.

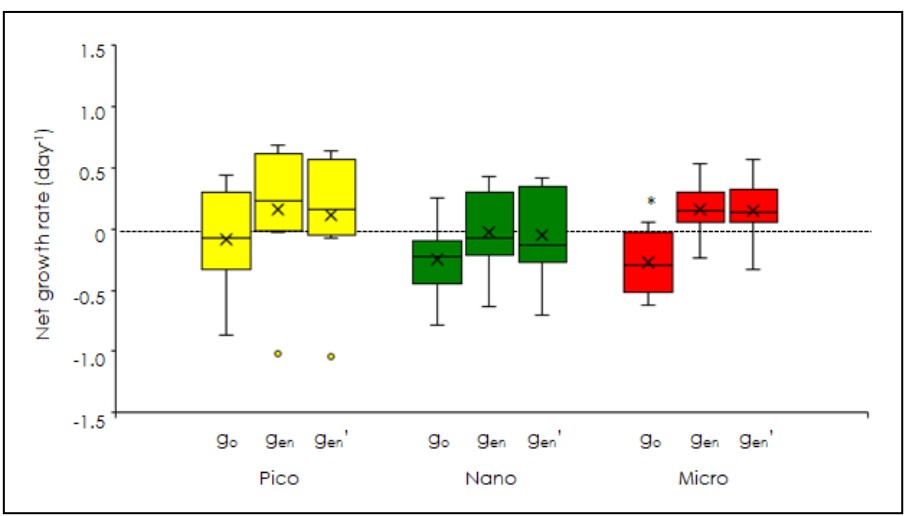


**Figure 7** Comparisons of phytoplankton net growth derived from the dilution experiments (EXP$_b$) among the three different methods. $g_o$: Observed net growth rates without enrichment in the non-diluted bottles. $g_{en}$: Observed net growth rates with enrichment in the non-diluted bottles. $g_{en}'$: Estimated net growth rates subtracting the mortality rates ($m$) from the maximum growth rates ($g_{max}$). Box-and-whisker diagram at each nitrate concentration was compiled from the results conducted at the 8 stations. Asterisk means significant difference between $g_o$ and $g_{en}$ (Welch's $t$-test, $p < 0.05$). The symbols have the same meaning as in Figure 6.

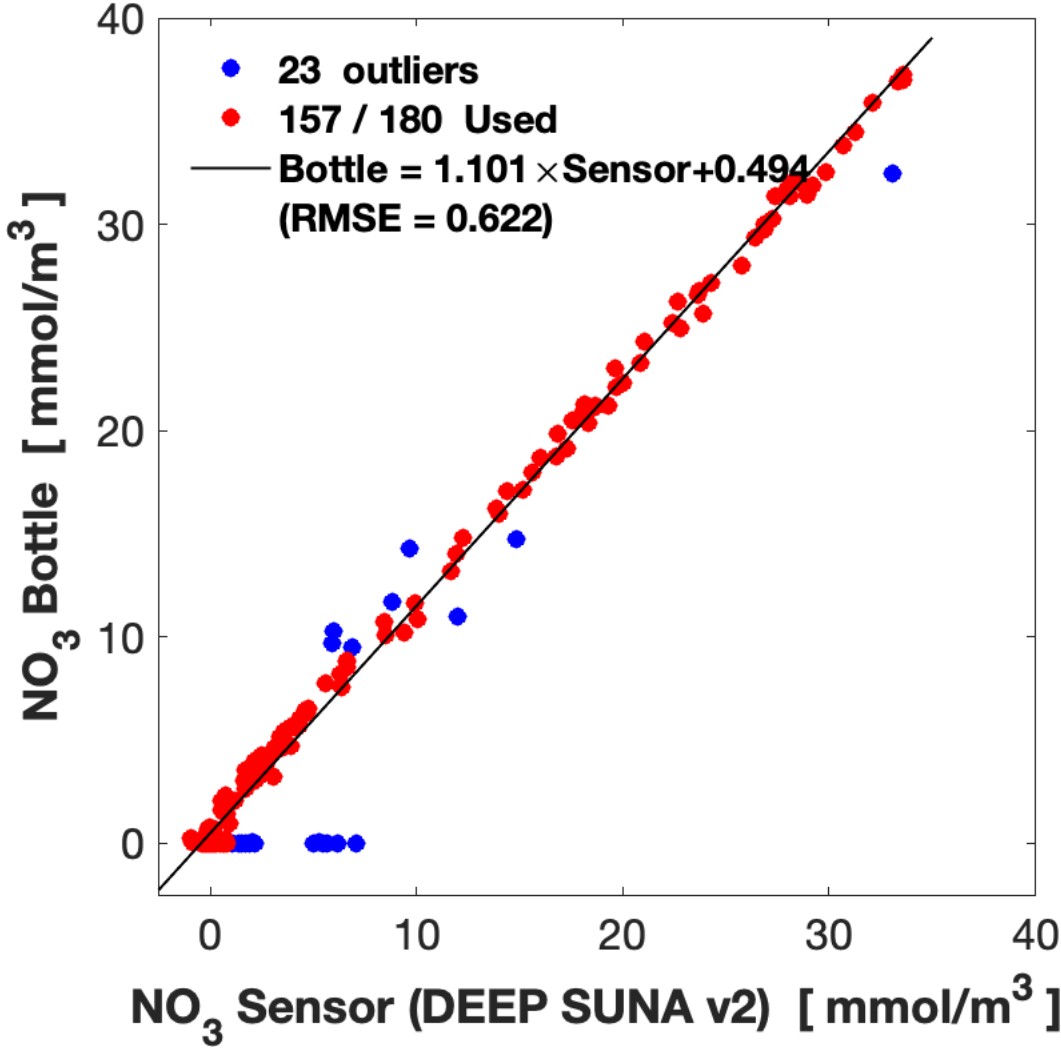

519
520

**Supplement Figure 1** In situ nitrate measurements by Deep SUNA V2 plotted against the laboratory water analysis
results from bottle sampled water during cruise KG1515 of T/S *Kagoshima-maru*. For obtaining the regression line used
for the sensor calibration, we excluded outlier data in which the absolute value of the difference between the data and
regression line exceeded 2.2 times the RMSE.


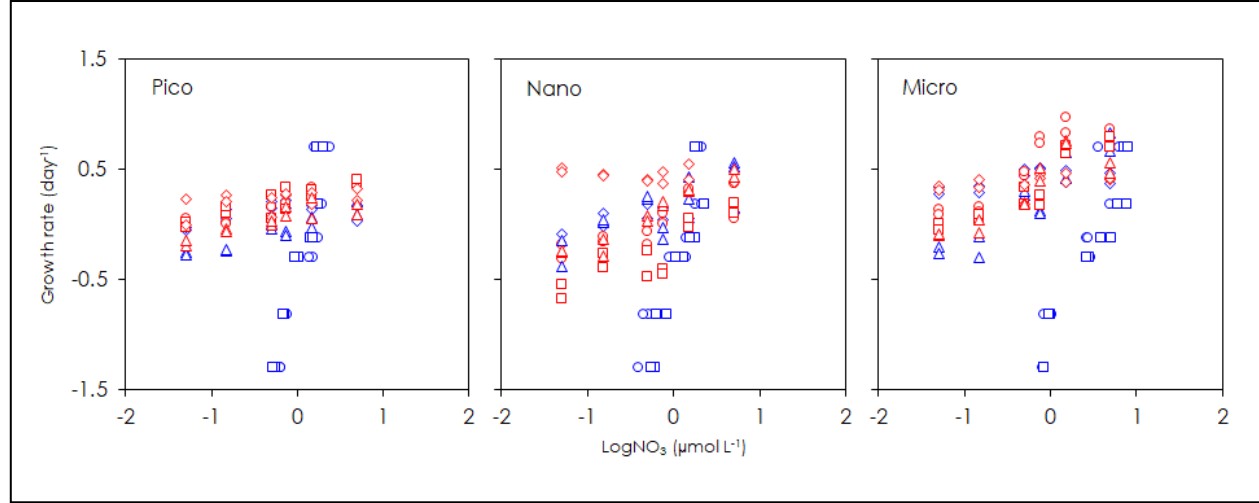


**Supplement Figure 2** Relationship of phytoplankton growth rates to logarithmically transformed concentrations of
enriched nitrate. Blue and red circles mean the stations in the upstream and downstream Kuroshio in the Tokara Strait,
respectively. Pico: chlorophyll smaller than 2 μm. Nano: chlorophyll between 2 and 11 μm. Micro: chlorophyll larger
than 11 μm.