# Peer review of "Phytoplankton growth and consumption by microzooplankton stimulated by turbulent nitrate flux suggest rapid trophic transfer in the oligotrophic Kuroshio"

_Biogeosciences, 2019_

## Referee Comment (RC1) · Chih-hao Hsieh (Referee) · 29 Oct 2019

Review Kobari et al 2019 Biogeoscience
This study aims to demonstrate in the Kuroshio area near northern East China Sea, that turbulence-induced nitrate flux can stimulate phytoplankton production in this seemingly oligotrophic ocean, while microzooplankton respond quickly to graze down the phytoplankton. As a consequence, high phytoplankton biomass is not observable. The authors used turbulence and nitrate sensor to demonstrate the nitrate flux, use nutrient enrichment experiments to demonstrate effects of nitrate flux on phytoplankton growth, and dilution experiments to measure microzooplankton grazing. This work is really interesting and deserves publishing in Biogeosciences. I have following comments that aim to help improve this manuscript.

Main concerns:
1. Potential effects of microzooplankton:phytoplankton ratio on the enrichment experiments: Table 1 shows that the chl-*a* and microzooplankton standing stock at the beginning of the incubations varied. The relative abundance of microzooplankton to phytoplankton may change the strength of top-down control. I wonder if adding microzooplankton:chl-a ratio or standing stock of microzooplankton and chl-a density to the regression analysis (Figure 5) can further explain the variation of phytoplankton growth after enrichment.
2. Enrichment experiments that did not exhibit clear effect of ambient nutrient on phytoplankton growth enhancement to enrichment (Lines 161-167 and Figure 5): Indeed there is a negative trend between phytoplankton growth-enrichment regression slope and $[NO_3^-+NO_2^-]$ or $[PO_4^{3-}]$ in control experiments. However, the plankton communities that have small regression slopes and low $r^2$ ($r^2<0.5$; F01 and K08 in Fig. 5 and Table 1, which I labeled in the figure below) experienced quite different *in situ* nutrient condition, and only K08 seems to drive the negative trend. I would like to know if the negative trend remains after removing these two sets of low-$r^2$ points. Furthermore, is there any possible explanation why the two incubations under low and high nutrient concentration reacted similarly to nutrient enrichment?

[Figure]

3. "Intra-guild" predation within microzooplankton community (Line 158-160): The results indicate that enrichment slightly increased the growth rate of nauplii but not always increase ciliate growth, especially when enrichment is low. According to the biomass change of the three types of microzooplankton to enrichment, the increase of nauplii is not as significant as ciliates when enrichment is high (Figure 3). I think, maybe the intra-guild predation of ciliates by nauplii inhibit the growth of ciliates when ciliate growth enhanced by low enrichment was not strong enough to compensate their mortality by nauplii feeding. As the enrichment increase further, fast growing ciliates can outgrow the consumption by large nauplii that grow and react more slowly to environmental change,

and thus ciliate growth and biomass accumulation increase. If the body size ratio between nauplii and ciliates in the incubations fit the predator-prey mass ratio of nauplii (Hansen et al. 1994), this is possible to happen.

4.  Stoichiometry of nutrient supply in Kuroshio (Lines 82-83): The enrichment and dilution experiments supplied phytoplankton with nitrate and phosphate molar concentration in 15:1 ratio (slightly N-limited, relative to the Redfield ratio 16:1). Did this ratio mimic the inorganic N:P concentration ratio or N:P flux by turbulent mixing in Kuroshio? Since this study focus on the nitrate supply from turbulent mixing, I expect that N should be limited. Nevertheless, I would like to know more about the stoichiometric condition of this study area and its potential effect on phytoplankton growth.

5.  I will appreciate data to demonstrate the accuracy of in situ nitrate sensor (e.g. comparing with measurements using water collected by sampling bottles). This issue is particularly important when nitrate concentration is low in the water.

6.  English needs substantial polishing to ensure correct grammar and wording. Some sentences are difficult to understand.

Editorial comments:
Abstract:
I have concerns on "rapid trophic transfer" in the title. The authors show evidence of rapid microzooplankton consumption of phytoplankton, but did not show evidence of trophic transfer.
Suggested title:
"Phytoplankton growth and consumption by microzooplankotn stimulated by turbulent nitrate flux suggest rapid trophic transfer in the oligotrophic Kuroshio

The writing of Abstract is confusing. Readers cannot tell what are the results obtained from the experiments, what are the results from other studies, and what are the inferences from those results. I think these issues need to be clearly clarified in Abstract.

Line 29: I cannot understand this sentence, and what the authors intend to say.

Line 31: This conclusion sentence is inference based on the results and should be written as so.

Line 35: "were simulated"

Line 35: "Results of dilution …

Line 37: Please explain what you mean by "invisible".

Introduction
Line 40: I cannot understand what is "originates to".

Line 43: In spite of such "seemingly" unproductive

Line 46: I cannot understand this sentence.

Methods:

Line 78: Please explain the motivation of using nutrient gradient in experiment in this paragraph, so that the readers can follow the logic flow better.

Typically in dilution exp, nutrients were amended in all bottles of the 4 dilution factors. Then, to evaluate whether nutrient limitation exists, additional no nutrient amended exp is conducted for non-diluted bottles (100%). Is this the protocol in the EXPb? Please clarify. If the authors did not follow this protocol, please explain why.

Line 100: Please explain how the chla data from different size fraction was obtained in this section.

Line 120: Please clarify the difference between the $C_t$ in equation (2) and (3). The explanation is confusing.

Results
Line 131: confidence interval of "what"?

Line 136: what is "$O$"? I cannot understand this sentence.

Line 164: Is the "N concentration" the nitrate concentration in the control groups at the start of incubation, i.e. the nitrate concentration in the ambient seawater without enrichment?

Line 179: do you mean "$g_{en}'=g_{max}-m$"?

Line 184: Do you mean $g_{en}$ here when referring to net growth rate?

Discussion
Line 191: should be ""previous", not previously

Line 225: This sentence is confusing. Previous sentence said that "microzooplankton standing stocks and growths are not elevated".

Line 235: Because microzooplankton growth rate and standing stocks are NOT significantly elevated, I am NOT sure that the authors can conclude the "rapidly transferred to microzooplankton via their grazing".

Figures:
Figure 2a: The unit of the orange curve seems to be the vertical gradient of nitrate, not the concentration. Please confirm whether this is the concentration or gradient curve.

Figure 3b and 4b: Please use a different set of colors or shading to present the microzooplankton data. It is a little bit difficult to recognize the difference between subplots *a* and *b* in these two figures.

Figure 5: The color used to present the r values should be consistent to the color used in Figure 3, 4, and 6 (micro = red, nano = green, and pico = yellow). I found that the colors of the points used in this figure correspond to the right size classes but colors of the captions on

this figure seem not (micro = green, nano = red, pico = black).

---

## Author Comment (AC1) · 4 Nov 2019

Reply to comments

This study aims to demonstrate in the Kuroshio area near northern East China Sea, that turbulence-induced nitrate flux can stimulate phytoplankton production in this seemingly oligotrophic ocean, while microzooplankton respond quickly to graze down the phytoplankton. As a consequence, high phytoplankton biomass is not observable. The authors used turbulence and nitrate sensor to demonstrate the nitrate flux, use nutrient enrichment experiments to demonstrate effects of nitrate flux on phytoplankton growth, and dilution experiments to measure microzooplankton grazing. This work is really interesting and deserves publishing in Biogeosciences. I have following comments that aim to help improve this manuscript.

>We really appreciate your kind comments to our findings. We indicated point-by-point response to the following comments. Hopefully, these are enough responses to your comments and suggestions.

Main concerns:

1. Potential effects of microzooplankton:phytoplankton ratio on the enrichment experiments: Table 1 shows that the chl-a and microzooplankton standing stock at the beginning of the incubations varied. The relative abundance of microzooplankton to phytoplankton may change the strength of top-down control. I wonder if adding microzooplankton:chl-a ratio or standing stock of microzooplankton and chl-a density to the regression analysis (Figure 5) can further explain the variation of phytoplankton growth after enrichment.

   >Your comments are really great and thanks. As you mentioned, we computed correlation between the slope of phytoplankton growth rates to the nutrients gradients and micro-sized heterotrophs biomass. Because no significant correlation was found for any size fractions to micro-sized heterotrophs, we have deleted these results from the manuscript. However, since some readers might have similar point of your view, we added these results in Figure 5 and some descriptions in the revised manuscript as follows.

   "The slope of a linear regression between growth rates of the size-fractionated chlorophyll and the logarithms of the nitrate enrichments at each incubation provided a metric of the sensitivity of their growth rates to nutrient supply (Supplement Fig 1). To explain why growth rates of the size-fractionated chlorophyll varied among stations, the slopes were compared to the nitrate+nitrite (Fig 5a) and phosphate concentrations (Fig 5b) and microzooplankton biomass (Fig 5c) in the ambient seawater without enrichment. No significant correlation was found for all size-fractionated chlorophyll to the micro-sized heterotrophs biomass. On the other hand, there was a negative correlation of the slopes for all size-fractions to the nitrate plus nitrite or phosphate concentrations, indicating that the stimulation of their growth rates by nutrients supply was greater for all size-fractionated chlorophyll under more oligotrophic conditions."

[Figure]

Fig 5. (Kobari et al.)

2.  Enrichment experiments that did not exhibit clear effect of ambient nutrient on phytoplankton growth enhancement to enrichment (Lines 161-167 and Figure 5): Indeed there is a negative trend between phytoplankton growth-enrichment regression slope and [NO3−+NO2−] or [PO43−] in control experiments. However, the plankton communities that have small regression slopes and low r2 (r2<0.5; F01 and K08 in Fig. 5 and Table 1, which I labeled in the figure below) experienced quite different in situ nutrient condition, and only K08 seems to drive the negative trend. I would like to know if the negative trend remains after removing these two sets of low-r2 points. Furthermore, is there any possible explanation why the two incubations under low and high nutrient concentration reacted similarly to nutrient enrichment?

    >We appreciate nice comments. As you suggested, correlation coefficients are much low (−0.014 to −0.778) and no significant if the slope of the phytoplankton growth rates at both stations are removed from Figure 5. According to the results in Table 2, phytoplankton growths at both stations tended to be higher than those at the other stations and positive even under no enrichment, particularly for micro-sized phytoplankton. We reported that larger phytoplankton predominated in coastal waters were often entrapped in frontal eddies and meanders of the Kuroshio around the study sites and advected into the Kuroshio (Kobari et al. 2019, Geophysical Monograph 243: 223-243). Probable explanations are that growths of phytoplankton communities at both stations are already stimulated with the advected coastal waters before our bottle experiments and nutrients are consumed for those phytoplankton communities particularly at K08. There is no evidence to support such hypothesis, however, we could not add further explanations in the revised manuscript.

3. "Intra-guild" predation within microzooplankton community (Line 158-160): The results indicate that enrichment slightly increased the growth rate of nauplii but not always increase ciliate growth, especially when enrichment is low. According to the biomass change of the three types of microzooplankton to enrichment, the increase of nauplii is not as significant as ciliates when enrichment is high (Figure 3). I think, maybe the intraguild predation of ciliates by nauplii inhibit the growth of ciliates when ciliate growth enhanced by low enrichment was not strong enough to compensate their mortality by nauplii feeding. As the enrichment increase further, fast growing ciliates can outgrow the consumption by large nauplii that grow and react more slowly to environmental change, and thus ciliate growth and biomass accumulation increase. If the body size ratio between nauplii and ciliates in the incubations fit the predator-prey mass ratio of nauplii (Hansen et al. 1994), this is possible to happen.

>We appreciate your kind suggestions to our results. It might be another possibility. Based on our data sets, the ratio of mean equivalent spherical diameter of body mass between copepod nauplii (88 μm) and naked ciliates (16 μm) was estimated to be 5:1 and much different from to the predator-prey mass ratio (i.e., 18:1) reported by Hansen et al. (1994). As described above, no significant correlation was found for the growth response of phytoplankton to nutrients gradients. We think that such intraguild predation of copepod nauplii on naked ciliates would not happen in the bottles. However, we added such explanations in the revised manuscript as follows.

"On the other hand, "intra-guild" predation within micro-heterotrophs community might be another explanation on the less clear pattern of their standing stocks and growth rates. Growth rates of copepod nauplii were always higher than those of naked ciliates, especially under no or less nitrate supply. The ratio of mean equivalent spherical diameter of body mass between copepod nauplii (88 μm) and naked ciliates (16 μm) was estimated to be 5:1 and much different from to the predator-prey mass ratio (i.e., 18:1, Hansen et al., 1994). Thus, such intraguild predation of copepod nauplii on naked ciliates would not happen in the bottles. More importantly to no or less clear pattern of the growth of micro-heterotrophs, the results from the simultaneously conducted experiments imply that phytoplankton productivity is stimulated by the turbulent nitrate flux and rapidly grazed by microzooplankton but standing stocks and growths of micro-heterotrophs are not elevated during 3 days in the Kuroshio Current. Increase of micro-heterotrophs standing stocks and their trophic transfer to mesozooplankton might be found in the further downstream of the Kuroshio Current."

4. Stoichiometry of nutrient supply in Kuroshio (Lines 82-83): The enrichment and dilution experiments supplied phytoplankton with nitrate and phosphate molar concentration in 15:1 ratio (slightly N-limited, relative to the Redfield ratio 16:1). Did this ratio mimic the inorganic N:P concentration ratio or N:P flux by turbulent mixing in Kuroshio? Since this study focus on the nitrate supply from turbulent mixing, I expect that N should be limited. Nevertheless, I would like to know more about the stoichiometric condition of this study area and its potential effect on phytoplankton growth.

>You are right. As you can find in Table 1, the ratios of nitrate/nitrite and phosphate molar concentrations showed N-limited conditions at ambient waters excepted for one station. As reported by Hasegawa et al. (2019, Geophysical Monograph 243: 191-205), 15:1 was measured in the ECS-Kuroshio and defined for the stoichiometric ratio of nutrients enrichment in our bottle experiments. On the other hand, in my

knowledge, no information on the stoichiometric effects on phytoplankton growth is available in the ECS-Kuroshio.

5. I will appreciate data to demonstrate the accuracy of in situ nitrate sensor (e.g. comparing with measurements using water collected by sampling bottles). This issue is particularly important when nitrate concentration is low in the water.
   >Thanks. The measurement methodology for in situ nitrate sensor is just published in Japanese journal (Hasegawa et al. 2019, Bull Coast Oceanogr 27: 59-64). We added some explanation as follows referring the previous report.

   "The nitrate sensor was calibrated with the observed nitrate concentrations (accuracy: 0.37 mmol m-3, Hasegawa et al. 2019)."

6. English needs substantial polishing to ensure correct grammar and wording. Some sentences are difficult to understand.
   >Thanks for suggestion. We checked all phrases in the manuscript again and revised the incorrect grammars and words.

Editorial comments:

Abstract:

I have concerns on "rapid trophic transfer" in the title. The authors show evidence of rapid microzooplankton consumption of phytoplankton, but did not show evidence of trophic transfer. Suggested title: "Phytoplankton growth and consumption by microzooplankotn stimulated by turbulent nitrate flux suggest rapid trophic transfer in the oligotrophic Kuroshio

   >Yes, we agreed and revised the title as you suggested.

The writing of Abstract is confusing. Readers cannot tell what are the results obtained from the experiments, what are the results from other studies, and what are the inferences from those results. I think these issues need to be clearly clarified in Abstract.

Line 29: I cannot understand this sentence, and what the authors intend to say.
   >This sentence is revised as follows.

   "Even though vulnerable life stages of major foraging fishes have a risk to be entrapped by frontal eddies and meanders and encountered under the low food availability, they have life cycle strategies to grow and recruit around the Kuroshio Current."

Line 31: This conclusion sentence is inference based on the results and should be written as so.
   >Agree. We revised the phrase like this.

   "Here we report that phytoplankton growth and consumption by microzooplankton is stimulated by turbulent nitrate flux amplified with the Kuroshio Current."

Line 35: "were simulated"

>Thanks. We revised.

Line 35: "Results of dilution …
>Thanks. We added.

Line 37: Please explain what you mean by "invisible".
>Yes, we wanted to mean "phytoplankton and microzooplankton productivity have long been undetectable by satellite images and oceanographic observations". Since the readers might be confused, however, we deleted "invisible".

Introduction

Line 40: I cannot understand what is "originates to".
>Thanks. We revised this phrase like this.
"The Kuroshio enters the East China Sea from the east of Taiwan and flows along the continental slope until it passes through the Tokara Strait into the western North Pacific (Fig 1a)."

Line 43: In spite of such "seemingly" unproductive
>Yes, we added.

Line 46: I cannot understand this sentence.
>Thanks for comments. We revised as follows.
"Highly vulnerable early life stages of many foraging species have a risk to grow and recruit under the oligotrophic and unproductive waters in the ECS-Kuroshio (hereafter called the "Kuroshio Paradox": Saito, 2019), even if the warm temperatures of the Kuroshio Current could enhance cellular metabolic processes and then growth."

Methods:

Line 78: Please explain the motivation of using nutrient gradient in experiment in this paragraph, so that the readers can follow the logic flow better.
>We mentioned the motivation just before this sentence. However, as you suggested, we explained the motivations for EXPa and EXPb just before the section of "Experimental setup" as follows.
"Two different types of bottle incubations were performed in the present study. For phytoplankton and micro-heterotrophs growth rates in response to in situ nitrate influx by turbulent mixing, bottle incubations with nutrient gradients (EXPa) were conducted at 8 stations in November 2016 and 2017. For microzooplankton grazing on phytoplankton, the dilution experiments (EXPb) followed by Landry and Hasset (1982) were done at 8 stations in November 2017 (Fig 1b, Table 1)."

Typically in dilution exp, nutrients were amended in all bottles of the 4 dilution factors. Then, to evaluate whether nutrient limitation exists, additional no nutrient amended exp is conducted for non-diluted bottles (100%). Is this the protocol in the EXPb? Please clarify. If the authors did not follow this protocol, please

explain why.

>Non-diluted bottles without nutrients were made for EXPb due to comparisons of phytoplankton growths between enriched and non-enriched series. Thus, we revised the explanation on dilution experiments like this.

"For evaluating nutrient limitation on phytoplankton growth, no enrichment was conducted for triplicate non-diluted bottles (100%) for EXPb."

Line 100: Please explain how the chla data from different size fraction was obtained in this section.

>Thanks. We described the size fractions as follows.

"Size fractions were defined as Pico for chlorophyll smaller than 2 μm, Nano for chlorophyll between 2 and 11 μm and Micro for chlorophyll larger than 11 μm"

Line 120: Please clarify the difference between the Ct in equation (2) and (3). The explanation is confusing.

>Thanks for comments. We used C't and C'o for EXPb.

Results

Line 131: confidence interval of "what"?

>Thanks. We revised this phrase like this.

"We obtained 16 pairs of vertical profiles for turbulent diffusivity and nitrate concentrations and estimated the averages and 95 percent confidence intervals of the vertical profiles."

Line 136: what is "O"? I cannot understand this sentence.

>"$O$" means "order level". Such descriptions are likely common for physical oceanography.

Line 164: Is the "N concentration" the nitrate concentration in the control groups at the start of incubation, i.e. the nitrate concentration in the ambient seawater without enrichment?

>Yes, we changed "at the start of the incubations" into "in the ambient seawater without enrichment".

Line 179: do you mean "gen'=gmax-m"?

>Yes, we do. We did not change the phrase.

Line 184: Do you mean gen here when referring to net growth rate?

>Yes, we do.

Discussion

Line 191: should be ""previous", not previously

>We revised it. Thanks.

Line 225: This sentence is confusing. Previous sentence said that "microzooplankton standing stocks and growths are not elevated".

>We revised the phrase like this.

"Increase of micro-heterotrophs standing stocks and their trophic transfer to mesozooplankton might be found in the further downstream of the Kuroshio Current."

Line 235: Because microzooplankton growth rate and standing stocks are NOT significantly elevated, I am NOT sure that the authors can conclude the "rapidly transferred to microzooplankton via their grazing".

>Thanks for suggestion. We revised the phrase as follows.

"Our study has provided the first experimental evidence that phytoplankton standing stocks and growths are stimulated by turbulent nutrient fluxes and rapidly grazed by microzooplankton."

Figures:

Figure 2a: The unit of the orange curve seems to be the vertical gradient of nitrate, not the concentration. Please confirm whether this is the concentration or gradient curve.

>Yes, you are right. We revised "nitrate gradient curve" in the caption.

[Figure]

Fig 2. (Kobari et al.)

Figure 3b and 4b: Please use a different set of colors or shading to present the microzooplankton data. It is a little bit difficult to recognize the difference between subplots a and b in these two figures.

>Thanks for comments. We changed the colors.

[Figure]

Fig 3. (Kobari et al.)

[Figure]

Fig 4. (Kobari et al.)

Figure 5: The color used to present the r values should be consistent to the color used in Figure 3, 4, and 6 (micro = red, nano = green, and pico = yellow). I found that the colors of the points used in this figure correspond to the right size classes but colors of the captions on this figure seem not (micro = green, nano = red, pico = black).

>Thanks too. We used same colors among the figures.

[Figure]

Fig 5. (Kobari et al.)

---

## Referee Comment (RC2) · Zhiyu Liu (Referee) · 1 Dec 2019

This is an interesting study seeking to solve the so-called Kuroshio Paradox. As a physical oceanographer with expertise in small-scale ocean physics I am not in a position to comment on the biological part of this paper, but I do have fundamental concerns on the physics the authors employed in this study. First of all, turbulent diffusivity was not "measured", but rather estimated involving important physical assumptions, such as isotropy of small-scale (3D) turbulence for the estimation of the turbulent kinetic energy (TKE) dissipation rate from microscale velocity shear measurements, and the Osborn

formula (i.e., a local energy balance assuming constant mixing efficiency) for the estimation of diffusivity from the TKE dissipation rate. These and the procedures of data processing should be explained at least briefly in the manuscript. This is in particular necessary given the interdisciplinary nature of the work; the readers with different backgrounds should be able to well appreciate the foundations of the numbers that the authors use to support their points. Moreover, and more crucially, although it has been customary (in the biogeochemical literature particularly) to estimate diapycnal turbulent fluxes considering only the diffusive flux (i.e., equation (1) in the manuscript), it is now well recognized that this is fundamentally improper, because there is always a diapycnal advective flux associated with the diffusive flux. The physical reason is in fact quite straightforward, that is, diapycnal mixing induces fluxes not only of passive properties such as nutrients, but also of the buoyancy, so that the density of the water parcel is changed due to mixing, and thus a diapycnal advective velocity is induced. These ideas have in fact been rigorously elaborated by Trevor McDougall in 1980s (albeit apparently with insufficient attentions), and the biogeochemical implications have recently been explained by Du et al. (2017). It would be very interesting to see how the refined estimate would affect the authors' results.

References: 1. McDougall, T. J. (1984). The relative roles of diapycnal and isopycnal mixing on subsurface water mass conversion. Journal of Physical Oceanography, 14(10), 1577–1589. 2. McDougall, T. J. (1987). Thermobaricity, cabbeling, and water-mass conversion. Journal of Geophysical Research, 92(C5), 5448–5464. 3. Du, C., Liu, Z., Kao, S.-J., & Dai, M. (2017). Diapycnal fluxes of nutrients in an oligotrophic oceanic regime: The South China Sea. Geophysical Research Letters, 44, 11,510–11,518.

---

## Author Comment (AC2) · 7 Dec 2019

Reply to comments

This is an interesting study seeking to solve the so-called Kuroshio Paradox. As a physical oceanographer with expertise in small-scale ocean physics I am not in a position to comment on the biological part of this paper, but I do have fundamental concerns on the physics the authors employed in this study.

>We appreciate your point of view to our findings. Our point-by-point responses are as follow. Hopefully, these are enough responses to your comments.

First of all, turbulent diffusivity was not "measured", but rather estimated involving important physical assumptions, such as isotropy of small-scale (3D) turbulence for the estimation of the turbulent kinetic energy (TKE) dissipation rate from microscale velocity shear measurements, and the Osborn formula (i.e., a local energy balance assuming constant mixing efficiency) for the estimation of diffusivity from the TKE dissipation rate. These and the procedures of data processing should be explained at least briefly in the manuscript. This is in particular necessary given the interdisciplinary nature of the work; the readers with different backgrounds should be able to well appreciate the foundations of the numbers that the authors use to support their points.

>We appreciate your kind suggestions and agree with you. As you mentioned, we add some descriptions at the Materials and Methods section in the revised manuscript as follows.

"The nitrate profiles were measured by a nitrate sensor (Deep SUNA V2) attached on a SBE-9plus CTD system. The turbulence diffusivity was estimated from microstructure measurements by TurboMAP-L (JFE Advantech Co. Ltd.) based on Osborn (1980)'s formula, which were deployed instantly after each CTD cast for the same stations. The nitrate sensor was calibrated with the observed nitrate concentrations (accuracy: 0.37 mmol m$^{-3}$, Hasegawa et al. 2019). Total of sixteen nitrate and the turbulence diffusivity profiles obtained among the stations at KG1515 cruise by T/S Kagoshima-maru across the Kuroshio path were averaged, then the profiles of the gradient of the averaged nitrate, and the averaged turbulence diffusivity were multiplied for each depth to get the averaged turbulent nitrate fluxes. Both parameters were binned and averaged within 10-meter intervals. The vertical gradient of the averaged nitrate profile ($C_{NO3}$) and the averaged vertical diffusivity profile ($K_z$) were then multiplied at each depth ($z$) to estimate the area-averaged vertical turbulent nitrate flux ($F_{NO3}$) with the following equation:"

Moreover, and more crucially, although it has been customary (in the biogeochemical literature particularly) to estimate diapycnal turbulent fluxes considering only the diffusive flux (i.e., equation (1) in the manuscript), it is now well recognized that this is fundamentally improper, because there is always a diapycnal advective flux associated with the diffusive flux. The physical reason is in fact quite straightforward, that is, diapycnal mixing induces fluxes not only of passive properties such as nutrients, but also of the buoyancy, so that the density of the water parcel is changed due to mixing, and thus a diapycnal advective velocity is induced. These ideas have in fact been rigorously elaborated by Trevor McDougall in 1980s (albeit apparently with insufficient attentions), and the biogeochemical implications have recently been explained by Du et al. (2017). It would be very interesting to see how the refined estimate would affect the authors' results.

>Thank you for the useful comment on the diapycnal advection contribution. We think that the important nutrient flux in the present study is the one across the euphotic depth, not through the density layer, which is

transformed by the turbulent mixing. In addition, as our studied regions are frontal regions unlike the SCS, where the Kuroshio flows over the seamounts, density fluctuations should be caused not only by turbulent mixing but also by advection and the movement of the fronts. Accordingly, we focus our discussions on the vertical turbulent nutrient flux using cartesian coordinate, rather than diapycnal flux using isopycnal coordinate.

---

## Referee Comment (RC3) · Naomi Harada (Referee) · 18 Dec 2019

Phytoplankton productivity and rapid trophic transfer to microzooplankton stimulated by turbulent nitrate flux in oligotrophic Kuroshio Current by Kobari et al.

This manuscript suggests the potential mechanism to explain the biological richness (higher tropic level food web) of Kuroshio based on the indirect experimental results of cultured growth rate estimated by size fractionated Chl.a and mortality estimated by grazing pressure of microzooplankton. These indirect approaches are interesting and

might be valuable, however I think further explanation or evidences are necessary to make readers agree to the authors conclusion. I also agree with this manuscript for the possible publication in Biogeosciences after moderate revision. The substantial comments are as follows:

Introduction 1 The current version looks too simply. Why don't authors add the research background of this study citing references? For example, the importance of fish resources from Kuroshio is not described in this version and the significance of fish catch in the Kuroshio to the entire the North Pacific or global. In addition, what kind of lower trophic level organisms compose of assemblages of phytoplankton and zooplankton in the study area? What nutrient regulates the primary production in this study area N? or P? Etc. . .. 2 Nutrient supply mechanism by turbulent mixing or other physical processes should be more explained citing references, because there is a large gap between the paragraph 1 and 2 in the current introduction. 3 Why is Tokara Strait important in the Kuroshio track area? Is there any geographical characteristics or bottom topographic characteristics? Is the area of Tokara Strait hot spot of turbulent mixing? Is there any other hot spot of turbulent mixing in the Kuroshio track area? Please explain the above questions in the revised manuscript because the readers who are not familiar with Kuroshio and the North Pacific would not understand the significance of research of Tokara Strait.

Results 1 The manuscript described that nitrate flux induced by turbulent mixing at the subsurface Chl maximum was observed as 0.788 mmol m-2 d-1 in the Tokara Strait (150 km wide) and authors assumed that the same concentration was kept during 5 days. What potential physical mechanism does keep almost same nitrate concentration at the Chl maximum layer during week? 2 In terms of gradient enrichment experiment and dilution experiment, the please add further descriptions of the details e.g., methods themselves and what purpose are achieved by these methods etc. 3 Lines 161-167: I could not understand what authors would like to describe in this paragraph. Especially, the sentence of the line 163 (To explain . . .) seems quite to be abrupt. The more

explanation needs for Fig. 5. Does the fig 5 show the data comparing among all stations? Why can the Fig. 5 be used to explain the difference in growth rate of size fractionated Chl. a among stations? Please explain more details of the similarity or difference of characteristics among stations. In addition, no Supplement Fig.1 is attached in the manuscript.

Discussion 1 Line205: Why is microzooplankton standing stock in the Tokara Strait of the Kuroshio track low, although the grazing pressure of phytoplankton by microzooplankton are relatively large? Is there any evidence or previous studies to indicate the rapid energy transfer of the microzooplankton to larger size organisms? Please give the potential mechanism in the revised version. 2 Line219-220: The sentence of this line is abrupt because there is no evidence or discussion in terms of the large variation in microzooplankton standing stocks among stations.

---

## Author Comment (AC3) · 24 Dec 2019

Reply to comments

This manuscript suggests the potential mechanism to explain the biological richness (higher tropic level food web) of Kuroshio based on the indirect experimental results of cultured growth rate estimated by size fractionated Chl.a and mortality estimated by grazing pressure of microzooplankton. These indirect approaches are interesting and might be valuable, however I think further explanation or evidences are necessary to make readers agree to the authors conclusion. I also agree with this manuscript for the possible publication in Biogeosciences after moderate revision. The substantial comments are as follows:

>We appreciate your comments. Our point-by-point responses are indicated below. Hopefully, these are enough responses to your comments and suggestions. The revised manuscript is indicated as blue-colored characters and the revised phrases to your comments and suggestions are shown in highlighted in yellow. Please note that the revised phrases to the other reviewers' comments and suggestions might be included in the all phrases because we already responded to the other reviewers (original manuscript was already revised).

Introduction 1: The current version looks too simply. Why don't authors add the research background of this study citing references? For example, the importance of fish resources from Kuroshio is not described in this version and the significance of fish catch in the Kuroshio to the entire the North Pacific or global. In addition, what kind of lower trophic level organisms compose of assemblages of phytoplankton and zooplankton in the study area? What nutrient regulates the primary production in this study area N? or P? Etc....

>Thanks for suggestions. Just after this manuscript was submitted to Biogeosciences, the review papers have been published. Based on these results, we added more description on the research background.

In spite of such seemingly unproductive conditions, the Kuroshio in the East China Sea (ECS-Kuroshio) is neighboring major spawning and nursery grounds for foraging species such as sardine (Watanabe et al., 1996), jack mackerel (Sassa et al., 2008), and chub mackerel (Sassa and Tsukamoto, 2010), and common squid (Bower et al., 1999). Indeed, good fishing grounds have been formed for various fishes and squid using the Kuroshio and their catches composed more than half of total catch in Japan (Saito, 2019). Highly vulnerable early life stages of many foraging species have a risk to grow and recruit under the oligotrophic and unproductive waters in the ECS-Kuroshio (hereafter called the "Kuroshio Paradox": Saito, 2019), even if the warm temperatures of the Kuroshio Current could enhance cellular metabolic processes and then growth………Use of waters in the vicinity of the oligotrophic Kuroshio as a nursery and feeding ground would therefore appear to be a risky strategy unless there is a mechanism that enhance biological production in the Kuroshio.

There is increasing information on community structure of phyto- and zooplankton in the Kuroshio. Pico- to nano-autotrophs contributed to phytoplankton standing stocks in the Kuroshio and predominant components were cellular cyanobacteria like Prochlorococcus and Synechococcus, haptophytes and diatoms (Hasegawa et al. 2019; Endo and Suzuki 2019). Heterotrophic bacteria and calanoid copepods contributed to heterotrophs biomass in the Kuroshio, while microzooplankton biomass were minor (Kobari et al. 2019). Based on the mass balance model, mesozooplankton standing stocks were supported by micro- and nano-autotrophs and microzooplankton (Kobari et al. 2019). However, we

have little knowledge how biogeochemical processes and trophodynamics support plankton community in the Kuroshio.

Here we report phytoplankton productivity and subsequent microzooplankton grazing stimulated by turbulent nitrate flux that can happen in the Kuroshio Current.

Introduction 2: Nutrient supply mechanism by turbulent mixing or other physical processes should be more explained citing references because there is a large gap between the paragraph 1 and 2 in the current introduction.

Introduction 3: Why is Tokara Strait important in the Kuroshio track area? Is there any geographical characteristics or bottom topographic characteristics? Is the area of Tokara Strait hot spot of turbulent mixing? Is there any other hot spot of turbulent mixing in the Kuroshio track area? Please explain the above questions in the revised manuscript because the readers who are not familiar with Kuroshio and the North Pacific would not understand the significance of research of Tokara Strait.

>We appreciate your kind suggestions. Agree. The two issues you suggested are associated each other. We added more description on the nutrients supply mechanisms and importance of the Tokara Strait before the last paragraph in Introduction section. The information was also based on the recent review papers as mentioned above.

In recent years, some mechanisms have been found for nutrients supply to the oligotrophic Kuroshio waters. The Kuroshio nutrient stream contributed significantly to productivity in the euphotic layer, similarly to the "nutrient stream" along the Gulf Stream (Komatsu and Hiroe, 2019). Turbulence around the Kuroshio appeared to be important for upward nutrients supply in the Kuroshio (Nagai et al., 2019). Frontal disturbances also contributed to nutrients supply to the surface layer in the Kuroshio (Kuroda, 2019). Moreover, the Island Mass Effect was produced by the Kuroshio Current around the archipelagic topography and induced upward nutrients supply (Hasegawa, 2019). These nutrients supplies have been suggested to stimulate biological productivity in the Kuroshio. In the wide Kuroshio track area, these nutrients supplies can happen particularly around the Tokara Straits due to the extensive frontal disturbances (Nakamura et al., 2006) and strong turbulence (Tsutsumi et al., 2017; Nagai et al., 2017, 2019).

Here we report phytoplankton productivity and subsequent microzooplankton grazing stimulated by turbulent nitrate flux that can happen in the Kuroshio Current.

Results 1: The manuscript described that nitrate flux induced by turbulent mixing at the subsurface Chl maximum was observed as 0.788 mmol m-2 d-1 in the Tokara Strait (150 km wide) and authors assumed that the same concentration was kept during 5 days. What potential physical mechanism does keep almost same nitrate concentration at the Chl maximum layer during week?

>Thanks for good comments. Our assumptions are based on the direct observations of turbulence (see Tsutsumi et al., 2017; Nagai et al., 2017). The strong turbulence was likely kept when the Kuroshio Current passed over the Tokara Strait due to the narrow and shallow topography with many islands and seamount. Also, our assumption of the nitrate supply might be conservative in the ambient waters because the upward nutrients supplied with the Island Mass Effect was not considered here.

Results 2: In terms of gradient enrichment experiment and dilution experiment, the please add further descriptions of the details e.g., methods themselves and what purpose are achieved by these methods etc.

>Thanks for your suggestions. We mentioned them briefly at each paragraph but added clearer descriptions of the purpose and results achieved at the beginning and end of the phrases as follows.

Gradient enrichment experiments

To evaluate how these turbulent nitrate fluxes measured in the Tokara Strait increase the standing stocks of phytoplankton and micro-heterotrophs in the Kuroshio, we conducted bottle incubations of the phytoplankton and micro-heterotrophs communities enriched with the different nutrient concentrations (EXPa)…..Thus, the standing stocks of phytoplankton and micro-heterotrophs were likely increased within the range of the turbulent nitrate fluxes measured in the Tokara Strait.

To explain whether growth rates of the size-fractionated chlorophyll might be variable with initial nutrients concentrations (bottom-up control) and predator biomasses (top-down control) at the beginning of the experiments, the slopes were compared to the nitrate+nitrite (Fig 5a) and phosphate concentrations (Fig 5b) and micro-heterotrophs biomass (Fig 5c) in the ambient seawater without enrichment….. Thus, the variations in phytoplankton growth rates are likely associated with nutrients concentrations at the beginning of the incubations.

Dilution experiments

To evaluate how much and which size-fractionated phytoplankton was removed by microzooplankton grazing, the dilution experiments were conducted simultaneously to the gradient enrichment experiments…..These results imply that gen of all size-fractionated chlorophyll balances the microzooplankton grazing mortality with the maximum growth. Particularly for the nano-fractionated chlorophyll, the net growth rates were slightly low due to the mortality rates by microzooplankton grazing exceeded the maximum growth rates.

Results 3. Lines 161-167: I could not understand what authors would like to describe in this paragraph. Especially, the sentence of the line 163 (To explain ...) seems quite to be abrupt. The more explanation needs for Fig. 5. Does the fig 5 show the data comparing among all stations? Why can the Fig. 5 be used to explain the difference in growth rate of size fractionated Chl. a among stations? Please explain more details of the similarity or difference of characteristics among stations. In addition, no Supplement Fig.1 is attached in the manuscript.

>Thanks for your comments. We added more descriptions on the reason why we compared the slope of a linear regression of phytoplankton growths to nutrients supply using supplement Fig. 1 as follows. At the platform of Biogeosciences, supplement materials seem to be provided with different files from the manuscript. You can find the Supplement (205KB) below the manuscript PDF or XML files at the website.

The slope of a linear regression between growth rates of the size-fractionated chlorophyll and the logarithms of the nitrate enrichments at each incubation provided a metric of the sensitivity of their growth rates to nutrient supply. As shown in Supplement Fig 1, the steeper slopes were found at some

stations in the upstream Kuroshio in the Tokara Strait compared with those at the other stations, suggesting that apparent phytoplankton growths were variable with the nutrients concentrations or predatory impacts at the beginning of the incubations. To explain whether growth rates of the size-fractionated chlorophyll might be variable with initial nutrients concentrations (bottom-up control) or predator biomasses (top-down control) at the beginning of the experiments, the slopes were compared to the nitrate+nitrite (Fig 5a) and phosphate concentrations (Fig 5b) and micro-heterotrophs biomass (Fig 5c) in the ambient seawater without enrichment.

Discussion 1: Line205: Why is microzooplankton standing stock in the Tokara Strait of the Kuroshio track low, although the grazing pressure of phytoplankton by microzooplankton are relatively large? Is there any evidence or previous studies to indicate the rapid energy transfer of the microzooplankton to larger size organisms? Please give the potential mechanism in the revised version.

>Thanks for your comments to the low microzooplankton biomass. Unfortunately, there is no direct evidence why microzooplankton biomass was low in the Kuroshio, excepted for the indirect evidence that microzooplankton might be removed by mesozooplankton predation based on the carbon flow among various components (Kobari et al., 2019). Thus, we added this brief information there.

Microzooplankton standing stocks in the Kuroshio Current at the Tokara Strait were lower than those on the continental shelf of the ECS (Chen et al., 2003) and might be removed by mesozooplankton predation (Kobari et al., 2019). These results expected low microzooplankton grazing on phytoplankton.

On the other hand, we have conducted the other bottle experiments to evaluate how much microzooplankton was removed by mesozooplankton predations. As you expected, the results from the bottle experiments demonstrated that naked ciliates predominated in microzooplankton biomass were removed by mesozooplankton predation. Since these results are recently submitted, we could not mention more.

Discussion 2: Line219-220: The sentence of this line is abrupt because there is no evidence or discussion in terms of the large variation in microzooplankton standing stocks among stations.

>Thanks. Large variations in microzooplankton standing stocks among the stations were already shown in Table 1, and thus we added "Table 1" in this sentence.

---

## Author Response (AR1)

**Reply to RC1**

This study aims to demonstrate in the Kuroshio area near northern East China Sea, that turbulence-induced nitrate flux can stimulate phytoplankton production in this seemingly oligotrophic ocean, while microzooplankton respond quickly to graze down the phytoplankton. As a consequence, high phytoplankton biomass is not observable. The authors used turbulence and nitrate sensor to demonstrate the nitrate flux, use nutrient enrichment experiments to demonstrate effects of nitrate flux on phytoplankton growth, and dilution experiments to measure microzooplankton grazing. This work is really interesting and deserves publishing in Biogeosciences. I have following comments that aim to help improve this manuscript.

>We appreciate your kind comments to our findings. As shown in BGD, we indicated point-by-point response to the following comments. Some responses to RC1 at the last time (BGD) were little changed after receiving the RC2 and RC3, but the revised phrases are substantially same. Hopefully, these are enough responses to your comments and suggestions.

**Main concerns:**

Potential effects of microzooplankton:phytoplankton ratio on the enrichment experiments: Table 1 shows
that the chl-a and microzooplankton standing stock at the beginning of the incubations varied. The relative
abundance of microzooplankton to phytoplankton may change the strength of top-down control. I wonder
if adding microzooplankton:chl-a ratio or standing stock of microzooplankton and chl-a density to the
regression analysis (Figure 5) can further explain the variation of phytoplankton growth after enrichment.
>As we mentioned at BGD, we computed correlation between the slope of phytoplankton growth rates to
the nutrients gradients and micro-sized heterotrophs, we have deleted these results from the manuscript.
However, since some readers might have similar point of view, we added these results in Figure 5 and
some descriptions in the revised manuscript as follows (L205-217).

"The slope of a linear regression between growth rates of the size-fractionated chlorophyll and the logarithms of the nitrate enrichments at each incubation provided a metric of the sensitivity of their growth rates to nutrient supply. As shown in Supplement Fig 1, the steeper slopes were found at some stations in the upstream Kuroshio in the Tokara Strait compared with those at the other stations, suggesting that apparent phytoplankton growths were variable with the nutrients concentrations or predatory impacts at the beginning of the incubations. To explain whether growth rates of the sizefractionated chlorophyll might be variable with initial nutrients concentrations (bottom-up control) or predator biomasses (top-down control) at the beginning of the experiments, the slopes were compared to the nitrate+nitrite (Fig 5a) and phosphate concentrations (Fig 5b) and microheterotrophs biomass (Fig 5c) in the ambient seawater without enrichment. No significant correlation was found for all size-fractionated chlorophyll to the micro-sized heterotrophs biomass. On the other hand, there was a negative correlation of the slopes for all size-fractions to the nitrate plus nitrite or phosphate concentrations, indicating that the stimulation of their growth rates by nutrients supply was greater for all size-fractionated chlorophyll under more oligotrophic conditions. Thus, the variations in phytoplankton growth rates are likely associated with nutrients concentrations at the beginning of the incubations."

2. Enrichment experiments that did not exhibit clear effect of ambient nutrient on phytoplankton growth enhancement to enrichment (Lines 161-167 and Figure 5): Indeed there is a negative trend between phytoplankton growth-enrichment regression slope and [NO3-+NO2-] or [PO43-] in control experiments. However, the plankton communities that have small regression slopes and low r2 (r2<0.5; F01 and K08 in Fig. 5 and Table 1, which I labeled in the figure below) experienced quite different in situ nutrient condition, and only K08 seems to drive the negative trend. I would like to know if the negative trend remains after removing these two sets of low-r2 points. Furthermore, is there any possible explanation why the two incubations under low and high nutrient concentration reacted similarly to nutrient enrichment?

>As you suggested, correlation coefficients are much low (-0.014 to -0.778) and no significant if the slope of the phytoplankton growth rates at both stations are removed from Figure 5. According to the results in Table 2, phytoplankton growths at both stations tended to be higher than those at the other stations and positive even under no enrichment, particularly for micro-sized phytoplankton. We reported that larger phytoplankton predominated in coastal waters were often entrapped in frontal eddies and meanders of the Kuroshio around the study sites and advected into the Kuroshio (Kobari et al. 2019, Geophysical Monograph 243: 223-243). Probable explanations are that growths of phytoplankton communities at both stations are already stimulated with the advected coastal waters before our bottle experiments and nutrients are consumed for those phytoplankton communities particularly at K08. There is no evidence to support such hypothesis, however, we could not add further explanations in the revised manuscript.

3. "Intra-guild" predation within microzooplankton community (Line 158-160): The results indicate that enrichment slightly increased the growth rate of nauplii but not always increase ciliate growth, especially when enrichment is low. According to the biomass change of the three types of microzooplankton to enrichment, the increase of nauplii is not as significant as ciliates when enrichment is high (Figure 3). I think, maybe the intraguild predation of ciliates by nauplii inhibit the growth of ciliates when ciliate growth enhanced by low enrichment was not strong enough to compensate their mortality by nauplii feeding. As the enrichment increase further, fast growing ciliates can outgrow the consumption by large nauplii that grow and react more slowly to environmental change, and thus ciliate growth and biomass accumulation increase. If the body size ratio between nauplii and ciliates in the incubations fit the predator-prey mass ratio of nauplii (Hansen et al. 1994), this is possible to happen.

>This might be another possibility. Based on our data sets, the ratio of mean equivalent spherical diameter of body mass between copepod nauplii (88  $\mu$ m) and naked ciliates (16  $\mu$ m) was estimated to be 5:1 and much different from to the predator-prey mass ratio (i.e., 18:1) reported by Hansen et al. (1994). As described above, no significant correlation was found for the growth response of phytoplankton to nutrients gradients. We think that such intraguild predation of copepod nauplii on naked ciliates would not happen in the bottles. However, we added such explanations in the revised manuscript as follows (L273-L283).

"On the other hand, "intra-guild" predation within micro-heterotrophs community might be another explanation on the less clear pattern of their standing stocks and growth rates. Growth rates of copepod nauplii were always higher than those of naked ciliates, especially under no or less nitrate supply. The ratio of mean equivalent spherical diameter of body mass between copepod nauplii (88 µm) and naked

ciliates (16  $\mu$ m) was estimated to be 5:1 and much different from to the predator-prey mass ratio (i.e., 18:1, Hansen et al., 1994). Thus, such intraguild predation of copepod nauplii on naked ciliates would not happen in the bottles. More importantly to no or less clear pattern of the growth of micro-heterotrophs, the results from the simultaneously conducted experiments imply that phytoplankton productivity is stimulated by the turbulent nitrate flux and rapidly grazed by microzooplankton but standing stocks and growths of micro-heterotrophs are not elevated during 3 days in the Kuroshio Current."

4. Stoichiometry of nutrient supply in Kuroshio (Lines 82-83): The enrichment and dilution experiments supplied phytoplankton with nitrate and phosphate molar concentration in 15:1 ratio (slightly N-limited, relative to the Redfield ratio 16:1). Did this ratio mimic the inorganic N:P concentration ratio or N:P flux by turbulent mixing in Kuroshio? Since this study focus on the nitrate supply from turbulent mixing, I expect that N should be limited. Nevertheless, I would like to know more about the stoichiometric condition of this study area and its potential effect on phytoplankton growth.

>You are right. As you can find in Table 1, the ratios of nitrate/nitrite and phosphate molar concentrations showed N-limited conditions at ambient waters excepted for one station. As reported by Hasegawa et al. (2019, Geophysical Monograph 243: 191-205), 15:1 was measured in the ECS-Kuroshio and defined for the stoichiometric ratio of nutrients enrichment in our bottle experiments. On the other hand, in my knowledge, no information on the stoichiometric effects on phytoplankton growth is available in the ECS-Kuroshio.

5. I will appreciate data to demonstrate the accuracy of in situ nitrate sensor (e.g. comparing with measurements using water collected by sampling bottles). This issue is particularly important when nitrate concentration is low in the water.

>The measurement methodology for in situ nitrate sensor is just published in Japanese journal (Hasegawa et al. 2019, Bull Coast Oceanogr 27: 59-64). We added detail explanation as follows referring the previous report (L86-110). We also demonstrated the supplement figure 1.

"The nitrate sensor was calibrated with the observed nitrate concentrations (Supplement Fig. 1). Since the precision of the nitrate sensor used in this study is low as 0.37 mmol m-3 (estimated by Hasegawa et. al., 2019), and the sampling rate (~2 samples m-1 for the sensor deployment speed of 0.5 m s-1) was coarse; if we calculate the vertical gradient from the raw data, the noise level would be too high for resolving the normal background nitrate stratification of O (10-1 mmol m-4). Therefore, need to set the vertical smoothing (averaging). Using the sensor value *Cs*, real value *Cr*, sensor precision *P* (0.37 mmol m-3), vertical deployment speed of sensor w, sampling frequency f and averaging bin size  $\Delta z$ , the bin averaged vertical gradient of sensor value can be written as

$$\frac{\partial \overline{Cs}}{\partial z} \sim \frac{\overline{Cr}_i - \overline{Cr}_{i-1}}{\Delta z} \pm P_{\sqrt{\frac{2\overline{w}}{\Delta z^3 f}}}$$
(1)

where, f = 1 Hz,  $\bar{w} = 0.5$  m s-1 in this study. The second term of the right side of Eq. (1) indicates the expected precision of the bin averaged vertical gradient of nitrate (see the detailed discussions in Hasegawa et. al., 2019). In this study, we took  $\Delta z = 10$  m to resolve the realistic vertical gradient with

the expected error size in  $O(10^{-2} \text{ mmol m}^{-4})$ . Total of sixteen nitrate and the turbulence diffusivity profiles obtained among the stations at KG1515 cruise by T/S Kagoshima-maru across the Kuroshio path were averaged, then the profiles of the gradient of the averaged nitrate, and the averaged turbulence diffusivity were multiplied for each depth to get the averaged turbulent nitrate fluxes. Both parameters were binned and averaged within 10-meter intervals. The vertical gradient of the averaged nitrate profile ( $C_{NO3}$ ) and the averaged vertical diffusivity profile ( $K_z$ ) were then multiplied at each depth (z) to estimate the area-averaged vertical turbulent nitrate flux ( $F_{NO3}$ ) with the following equation:

**$F_{NO3} = -K_Z \times \partial C_{NO3} / \partial z$**

In recent years, there is an active discussion about the importance of diapycnal advective flux associated with the diffusive flux (e.g., Du et al., 2017); however, in the present study, we assumed that the important nutrient flux was the one across the euphotic depth, not through the density layer, which was transformed by the turbulent mixing. In addition, as our studied regions were frontal regions unlike the South China Sea, where the Kuroshio flows over the seamounts, density fluctuations should be caused not only by turbulent mixing but also by advection and the movement of the fronts. Accordingly, we focus our discussions on the vertical turbulent nutrient flux using cartesian coordinate, rather than diapycnal flux using isopycnal coordinate."

(2)

6. English needs substantial polishing to ensure correct grammar and wording. Some sentences are difficult to understand.

>We checked all phrases in the manuscript again and revised the incorrect grammars and words.

**Editorial comments:**

**Abstract:**

I have concerns on "rapid trophic transfer" in the title. The authors show evidence of rapid microzooplankton consumption of phytoplankton, but did not show evidence of trophic transfer. Suggested title: "Phytoplankton growth and consumption by microzooplankotn stimulated by turbulent nitrate flux suggest rapid trophic transfer in the oligotrophic Kuroshio

>The title was revised as you suggested.

The writing of Abstract is confusing. Readers cannot tell what are the results obtained from the experiments, what are the results from other studies, and what are the inferences from those results. I think these issues need to be clearly clarified in Abstract.

Line 29: I cannot understand this sentence, and what the authors intend to say.

>This sentence is revised as follows (L28-30).

"Even though vulnerable life stages of major foraging fishes have a risk to be entrapped by frontal eddies and meanders and encountered under the low food availability, they have life cycle strategies to grow and recruit around the Kuroshio Current."

Line 31: This conclusion sentence is inference based on the results and should be written as so.

>We revised the phrase like this (L30-31).

"Here we report that phytoplankton growth and consumption by microzooplankton is stimulated by turbulent nitrate flux amplified with the Kuroshio Current."

```
Line 35: "were simulated"
```

>We revised (L34).

Line 35: "Results of dilution ...

>We added (L34).

Line 37: Please explain what you mean by "invisible".

>We wanted to mean "phytoplankton and microzooplankton productivity have long been undetectable by satellite images and oceanographic observations". Since the readers might be confused, however, we deleted this word, "invisible" (L36).

**Introduction**

Line 40: I cannot understand what is "originates to".

>We revised this phrase like this (L39-40).

"The Kuroshio enters the East China Sea from the east of Taiwan and flows along the continental slope until it passes through the Tokara Strait into the western North Pacific (Fig 1a)."

Line 43: In spite of such "seemingly" unproductive

>Yes, we added (L42).

**Line 46: I cannot understand this sentence.**

>We revised as follows (L46-49).

"Highly vulnerable early life stages of many foraging species have a risk to grow and recruit under the oligotrophic and unproductive waters in the ECS-Kuroshio (hereafter called the "Kuroshio Paradox": Saito, 2019), even if the warm temperatures of the Kuroshio Current could enhance cellular metabolic processes and then growth."

**Methods:**

Line 78: Please explain the motivation of using nutrient gradient in experiment in this paragraph, so that the readers can follow the logic flow better.

>We mentioned the motivation just before this sentence. However, as you suggested, we explained the motivations for EXPa and EXPb just before the section of "Experimental setup" as follows (L111-115).

"Two different types of bottle incubations were performed in the present study. For phytoplankton and micro-heterotrophs growth rates in response to in situ nitrate influx by turbulent mixing, bottle incubations with nutrient gradients (EXPa) were conducted at 8 stations in November 2016 and 2017. For microzooplankton grazing on phytoplankton, the dilution experiments (EXPb) followed by Landry and Hasset (1982) were done at 8 stations in November 2017 (Fig 1b, Table 1)."

Typically in dilution exp, nutrients were amended in all bottles of the 4 dilution factors. Then, to evaluate whether nutrient limitation exists, additional no nutrient amended exp is conducted for non-diluted bottles (100%). Is this the protocol in the EXPb? Please clarify. If the authors did not follow this protocol, please explain why.

>Non-diluted bottles without nutrients were made for EXPb due to comparisons of phytoplankton growths between enriched and non-enriched series. Thus, we revised the explanation on dilution experiments like this (L131-132).

"For evaluating nutrient limitation on phytoplankton growth, no enrichment was conducted for triplicate non-diluted bottles (100%) for EXPb."

Line 100: Please explain how the chla data from different size fraction was obtained in this section.

>We described the size fractions as follows (L145-146).

"Size fractions were defined as Pico for chlorophyll smaller than 2  $\mu$ m, Nano for chlorophyll between 2 and 11  $\mu$ m and Micro for chlorophyll larger than 11  $\mu$ m"

Line 120: Please clarify the difference between the Ct in equation (2) and (3). The explanation is confusing. >We used C't and C'o for EXPb (L161).

**Results**

Line 131: confidence interval of "what"?

>We revised this phrase like this (L172-173).

"We obtained 16 pairs of vertical profiles for turbulent diffusivity and nitrate concentrations and estimated the averages and 95 percent confidence intervals of the vertical profiles."

Line 136: what is "O"? I cannot understand this sentence.

>"O" means "order level". Such descriptions are likely common for physical oceanography (L176).

Line 164: Is the "N concentration" the nitrate concentration in the control groups at the start of incubation, i.e. the nitrate concentration in the ambient seawater without enrichment?

>Yes, we changed "at the start of the incubations" into "in the ambient seawater without enrichment" (L212).

Line 179: do you mean "gen'=gmax-m"?

>Yes, we do. We did not change the phrase.

Line 184: Do you mean gen here when referring to net growth rate?

>Yes, we do.

Discussion Line 191: should be ""previous", not previously

>We revised it (L241).

Line 225: This sentence is confusing. Previous sentence said that "microzooplankton standing stocks and growths are not elevated".

>We revised the phrase like this (L281-283).

"Increase of micro-heterotrophs standing stocks and their trophic transfer to mesozooplankton might be found in the further downstream of the Kuroshio Current."

Line 235: Because microzooplankton growth rate and standing stocks are NOT significantly elevated, I am NOT sure that the authors can conclude the "rapidly transferred to microzooplankton via their grazing".

>We revised the phrase as follows (L290-292).

"Our study has provided the first experimental evidence that phytoplankton standing stocks and growths are stimulated by turbulent nutrient fluxes and rapidly grazed by microzooplankton."

Figures:

Figure 2a: The unit of the orange curve seems to be the vertical gradient of nitrate, not the concentration. Please confirm whether this is the concentration or gradient curve.

>We revised "nitrate gradient curve" in the caption (L466).

Figure 3b and 4b: Please use a different set of colors or shading to present the microzooplankton data. It is a little bit difficult to recognize the difference between subplots a and b in these two figures.

>We changed the colors (see revised Figures 3b and 4b).

Figure 5: The color used to present the r values should be consistent to the color used in Figure 3, 4, and 6 (micro = red, nano = green, and pico = yellow). I found that the colors of the points used in this figure correspond to the right size classes but colors of the captions on this figure seem not (micro = green, nano = red, pico = black).

>We used same colors among the figures (see revised Figure 5).

**Reply to RC2**

This is an interesting study seeking to solve the so-called Kuroshio Paradox. As a physical oceanographer with expertise in small-scale ocean physics I am not in a position to comment on the biological part of this paper, but I do have fundamental concerns on the physics the authors employed in this study.

>We appreciate your kind comments to our findings. As shown in BGD, we indicated point-by-point response to the following comments. Some responses to RC2 at the last time (BGD) might be little changed after receiving the RC3 and editor comments, but the revised phrases are substantially same. Hopefully, these are enough responses to your comments and suggestions.

First of all, turbulent diffusivity was not "measured", but rather estimated involving important physical assumptions, such as isotropy of small-scale (3D) turbulence for the estimation of the turbulent kinetic energy (TKE) dissipation rate from microscale velocity shear measurements, and the Osborn formula (i.e., a local energy balance assuming constant mixing efficiency) for the estimation of diffusivity from the TKE dissipation rate. These and the procedures of data processing should be explained at least briefly in the manuscript. This is in particular necessary given the interdisciplinary nature of the work; the readers with different backgrounds should be able to well appreciate the foundations of the numbers that the authors use to support their points.

>As RC2 suggested, detail descriptions were added at the Materials and Methods section in the revised manuscript as follows (L86-103).

"The nitrate sensor was calibrated with the observed nitrate concentrations (Supplement Fig. 1). Since the precision of the nitrate sensor used in this study is low as 0.37 mmol m-3 (estimated by Hasegawa et. al., 2019), and the sampling rate (~2 samples m-1 for the sensor deployment speed of 0.5 m s-1) was coarse; if we calculate the vertical gradient from the raw data, the noise level would be too high for resolving the normal background nitrate stratification of O (10-1 mmol m-4). Therefore, need to set the vertical smoothing (averaging). Using the sensor value *Cs*, real value *Cr*, sensor precision *P* (0.37 mmol m-3), vertical deployment speed of sensor w, sampling frequency f and averaging bin size  $\Delta z$ , the bin averaged vertical gradient of sensor value can be written as

$$\frac{\partial \overline{Cs}}{\partial z} \sim \frac{\overline{Cr}_i - \overline{Cr}_{i-1}}{\Delta z} \pm P_{\sqrt{\Delta z^3 f}}$$
(1)

where, f = 1 Hz,  $\bar{w} = 0.5$  m s-1 in this study. The second term of the right side of Eq. (1) indicates the expected precision of the bin averaged vertical gradient of nitrate (see the detailed discussions in Hasegawa et. al., 2019). In this study, we took  $\Delta z = 10$  m to resolve the realistic vertical gradient with the expected error size in  $O(10^{-2} \text{ mmol m}^{-4})$ . Total of sixteen nitrate and the turbulence diffusivity profiles obtained among the stations at KG1515 cruise by T/S Kagoshima-maru across the Kuroshio path were averaged, then the profiles of the gradient of the averaged nitrate, and the averaged turbulence diffusivity were multiplied for each depth to get the averaged turbulent nitrate fluxes. Both parameters were binned and averaged within 10-meter intervals. The vertical gradient of the averaged nitrate profile ( $C_{NO3}$ ) and the averaged vertical diffusivity profile ( $K_2$ ) were then multiplied at each depth (z) to estimate the area-averaged vertical turbulent nitrate flux ( $F_{NO3}$ ) with the following equation:

$$F_{NO3} = -K_Z \times \partial C_{NO3} / \partial z$$

(2)"

Moreover, and more crucially, although it has been customary (in the biogeochemical literature particularly) to estimate diapycnal turbulent fluxes considering only the diffusive flux (i.e., equation (1) in the manuscript), it is now well recognized that this is fundamentally improper, because there is always a diapycnal advective flux associated with the diffusive flux. The physical reason is in fact quite straightforward, that is, diapycnal mixing induces fluxes not only of passive properties such as nutrients, but also of the buoyancy, so that the density of the water parcel is changed due to mixing, and thus a diapycnal advective velocity is induced. These ideas have in fact been rigorously elaborated by Trevor McDougall in 1980s (albeit apparently with insufficient attentions), and the biogeochemical implications have recently been explained by Du et al. (2017). It would be very interesting to see how the refined estimate would affect the authors' results.

>Brief explanations were added at the Materials and Methods section in the revised manuscript as follows (L104-110).

"In recent years, there is an active discussion about the importance of diapycnal advective flux associated with the diffusive flux (e.g., Du et al., 2017); however, in the present study, we assumed that the important nutrient flux was the one across the euphotic depth, not through the density layer, which was transformed by the turbulent mixing. In addition, as our studied regions were frontal regions unlike the South China Sea, where the Kuroshio flows over the seamounts, density fluctuations should be caused not only by turbulent mixing but also by advection and the movement of the fronts. Accordingly, we focus our discussions on the vertical turbulent nutrient flux using cartesian coordinate, rather than diapycnal flux using isopycnal coordinate."

This manuscript suggests the potential mechanism to explain the biological richness (higher tropic level food web) of Kuroshio based on the indirect experimental results of cultured growth rate estimated by size fractionated Chl.a and mortality estimated by grazing pressure of microzooplankton. These indirect approaches are interesting and might be valuable, however I think further explanation or evidences are necessary to make readers agree to the authors conclusion. I also agree with this manuscript for the possible publication in Biogeosciences after moderate revision. The substantial comments are as follows:

>As shown in BGD, we indicated point-by-point response to the following comments. Some responses to RC3 at the last time (BGD) might be little changed after receiving the editor comments, but the revised phrases are substantially same. Hopefully, these are enough responses to your comments and suggestions.

Introduction 1: The current version looks too simply. Why don't authors add the research background of this study citing references? For example, the importance of fish resources from Kuroshio is not described in this version and the significance of fish catch in the Kuroshio to the entire the North Pacific or global. In addition, what kind of lower trophic level organisms compose of assemblages of phytoplankton and zooplankton in the study area? What nutrient regulates the primary production in this study area N? or P? Etc....

>Thanks for suggestions. Just after this manuscript was submitted to Biogeosciences, the review papers have been published. Based on these results, we added more description on the research background. These revisions were highlighted in yellow (L42-62).

"In spite of such seemingly unproductive conditions, the Kuroshio in the East China Sea (ECS-Kuroshio) is neighboring major spawning and nursery grounds for foraging species such as sardine (Watanabe et al., 1996), jack mackerel (Sassa et al., 2008), and chub mackerel (Sassa and Tsukamoto, 2010), and common squid (Bower et al., 1999). Indeed, good fishing grounds have been formed for various fishes and squid using the Kuroshio and their catches composed more than half of total catch in Japan (Saito, 2019). Highly vulnerable early life stages of many foraging species have a risk to grow and recruit under the oligotrophic and unproductive waters in the ECS-Kuroshio (hereafter called the "Kuroshio Paradox": Saito, 2019), even if the warm temperatures of the Kuroshio Current could enhance cellular metabolic processes and then growth. It has been believed that survival of these early stages is supported by high plankton productivity on the continental shelf and in the Kuroshio front (Nakata et al., 1995). However, such good food availability is spatially limited and greatly variable because the Kuroshio Current often meanders (Nakata and Hidaka, 2003). Otherwise, the coastal water mass is sometimes entrapped and transported into the Kuroshio and more pelagic sites (Nakamura et al., 2006; Kobari et al., 2019). Use of waters in the vicinity of the oligotrophic Kuroshio as a nursery and feeding ground would therefore appear to be a risky strategy unless there is a mechanism that enhance biological production in the Kuroshio.

There is increasing information on community structure of phyto- and zooplankton in the Kuroshio. Pico- to nano-autotrophs contributed to phytoplankton standing stocks in the Kuroshio and predominant components were cellular cyanobacteria like Prochlorococcus and Synechococcus, haptophytes and diatoms (Hasegawa et al., 2019; Endo and Suzuki, 2019). Heterotrophic bacteria and calanoid copepods contributed to heterotrophs biomass in the Kuroshio, while microzooplankton biomass were minor (Kobari et al., 2019). Based on the mass balance model, mesozooplankton standing stocks were supported by micro- and nano-autotrophs and microzooplankton (Kobari et al., 2019). However, we have little knowledge how biogeochemical processes and trophodynamics support plankton community in the Kuroshio."

Introduction 2: Nutrient supply mechanism by turbulent mixing or other physical processes should be more explained citing references because there is a large gap between the paragraph 1 and 2 in the current introduction.

Introduction 3: Why is Tokara Strait important in the Kuroshio track area? Is there any geographical characteristics or bottom topographic characteristics? Is the area of Tokara Strait hot spot of turbulent mixing? Is there any other hot spot of turbulent mixing in the Kuroshio track area? Please explain the above questions in the revised manuscript because the readers who are not familiar with Kuroshio and the North Pacific would not understand the significance of research of Tokara Strait.

>The two issues are associated each other. We added more description on the nutrients supply mechanisms and importance of the Tokara Strait before the last paragraph in Introduction section. The information was also based on the recent review papers as mentioned above (L63-71).

"In recent years, some mechanisms have been found for nutrients supply to the oligotrophic Kuroshio waters. The Kuroshio nutrient stream contributed significantly to productivity in the euphotic layer, similarly to the "nutrient stream" along the Gulf Stream (Komatsu and Hiroe, 2019). Turbulence around the Kuroshio appeared to be important for upward nutrients supply in the Kuroshio (Nagai et al., 2019). Frontal disturbances also contributed to nutrients supply to the surface layer in the Kuroshio (Kuroda, 2019). Moreover, the Island Mass Effect was produced by the Kuroshio Current around the archipelagic topography and induced upward nutrients supply (Hasegawa, 2019). These nutrients supplies have been suggested to stimulate biological productivity in the Kuroshio. In the wide Kuroshio track area, these nutrients supplies can happen particularly around the Tokara Straits due to the extensive frontal disturbances (Nakamura et al., 2006) and strong turbulence (Tsutsumi et al., 2017; Nagai et al., 2017, 2019)."

Results 1: The manuscript described that nitrate flux induced by turbulent mixing at the subsurface Chl maximum was observed as 0.788 mmol m-2 d-1 in the Tokara Strait (150 km wide) and authors assumed that the same concentration was kept during 5 days. What potential physical mechanism does keep almost same nitrate concentration at the Chl maximum layer during week?

>Our assumptions are based on the direct observations of turbulence (see Tsutsumi et al., 2017; Nagai et al., 2017). The strong turbulence was likely kept when the Kuroshio Current passed over the Tokara Strait due to the narrow and shallow topography with many islands and seamount. Also, our assumption of the nitrate supply might be conservative in the ambient waters because the upward nutrients supplied with the Island Mass Effect was not considered here.

Results 2: In terms of gradient enrichment experiment and dilution experiment, the please add further

descriptions of the details e.g., methods themselves and what purpose are achieved by these methods etc.

>In the revised manuscript, we mentioned them briefly at each paragraph but added clearer descriptions of the purpose and results achieved at the beginning and end of the phrases as follows. Since the revised phrases are found everywhere, they are highlighted in yellow as follows.

Gradient enrichment experiments

L181-182

To evaluate how these turbulent nitrate fluxes measured in the Tokara Strait increase the standing stocks of phytoplankton and micro-heterotrophs in the Kuroshio, we conducted bottle incubations of the phytoplankton and micro-heterotrophs communities enriched with the different nutrient concentrations (EXPa).

L202-204

Thus, the standing stocks of phytoplankton and micro-heterotrophs were likely increased within the range of the turbulent nitrate fluxes measured in the Tokara Strait.

L209-211

To explain whether growth rates of the size-fractionated chlorophyll might be variable with initial nutrients concentrations (bottom-up control) and predator biomasses (top-down control) at the beginning of the experiments, the slopes were compared to the nitrate+nitrite (Fig 5a) and phosphate concentrations (Fig 5b) and micro-heterotrophs biomass (Fig 5c) in the ambient seawater without enrichment.

L216-217

Thus, the variations in phytoplankton growth rates are likely associated with nutrients concentrations at the beginning of the incubations.

**Dilution experiments**

L220-221

To evaluate how much and which size-fractionated phytoplankton was removed by microzooplankton grazing, the dilution experiments were conducted simultaneously to the gradient enrichment experiments.....These results imply that gen of all size-fractionated chlorophyll balances the microzooplankton grazing mortality with the maximum growth. Particularly for the nano-fractionated chlorophyll, the net growth rates were slightly low due to the mortality rates by microzooplankton grazing exceeded the maximum growth rates.

Results 3. Lines 161-167: I could not understand what authors would like to describe in this paragraph. Especially, the sentence of the line 163 (To explain ...) seems quite to be abrupt. The more explanation needs for Fig. 5. Does the fig 5 show the data comparing among all stations? Why can the Fig. 5 be used to explain the difference in growth rate of size fractionated Chl. a among stations? Please explain more details of the similarity or difference of characteristics among stations. In addition, no Supplement Fig.1 is attached in the manuscript.

>We added more descriptions on the reason why we compared the slope of a linear regression of phytoplankton growths to nutrients supply using supplement Fig. 1 as follows. At the platform of

Biogeosciences, supplement materials seem to be provided with different files from the manuscript. You can find the Supplement (205KB) below the manuscript PDF or XML files at the website (L205-212).

The slope of a linear regression between growth rates of the size-fractionated chlorophyll and the logarithms of the nitrate enrichments at each incubation provided a metric of the sensitivity of their growth rates to nutrient supply. As shown in Supplement Fig 1, the steeper slopes were found at some stations in the upstream Kuroshio in the Tokara Strait compared with those at the other stations, suggesting that apparent phytoplankton growths were variable with the nutrients concentrations or predatory impacts at the beginning of the incubations. To explain whether growth rates of the size-fractionated chlorophyll might be variable with initial nutrients concentrations (bottom-up control) or predator biomasses (top-down control) at the beginning of the experiments, the slopes were compared to the nitrate+nitrite (Fig 5a) and phosphate concentrations (Fig 5b) and micro-heterotrophs biomass (Fig 5c) in the ambient seawater without enrichment.

Discussion 1: Line205: Why is microzooplankton standing stock in the Tokara Strait of the Kuroshio track low, although the grazing pressure of phytoplankton by microzooplankton are relatively large? Is there any evidence or previous studies to indicate the rapid energy transfer of the microzooplankton to larger size organisms? Please give the potential mechanism in the revised version.

>Unfortunately, there is no direct evidence why microzooplankton biomass was low in the Kuroshio, excepted for the indirect evidence that microzooplankton might be removed by mesozooplankton predation based on the carbon flow among various components (Kobari et al., 2019). Thus, we added this brief information there (L255-257).

Microzooplankton standing stocks in the Kuroshio Current at the Tokara Strait were lower than those on the continental shelf of the ECS (Chen et al., 2003) and might be removed by mesozooplankton predation (Kobari et al., 2019). These results expected low microzooplankton grazing on phytoplankton. On the other hand, we have conducted the other bottle experiments to evaluate how much microzooplankton was removed by mesozooplankton predations. As you expected, the results from the bottle experiments demonstrated that naked ciliates predominated in microzooplankton biomass were removed by mesozooplankton predation. These results are recently submitted but could not be mentioned more here.

Discussion 2: Line219-220: The sentence of this line is abrupt because there is no evidence or discussion in terms of the large variation in microzooplankton standing stocks among stations (L262).

>Large variations in microzooplankton standing stocks among the stations were already shown in Table 1, and thus we added "Table 1" in this sentence.

[revised manuscript text omitted]

- $442 \qquad \text{chlorophyll smaller than 2 } \mu\text{m. Nano: chlorophyll between 2 and 11 } \mu\text{m. Micro: chlorophyll larger than 11 } \mu\text{m.}$

| Station | 0      |        | 0.05   |        | 0.15   |        | 0.5    |        | 0.75   |        | 1.5   |        | 5     |       |
|---------|--------|--------|--------|--------|--------|--------|--------|--------|--------|--------|-------|--------|-------|-------|
|         | A      | В      | А      | В      | A      | В      | А      | В      | A      | В      | A     | В      | A     | В     |
| Micro   |        |        |        |        |        |        |        |        |        |        |       |        |       |       |
| C02     | -0.108 | -0.116 | -0.089 | -0.082 | 0.019  | -0.073 | 0.470  | 0.426  | 0.422  | 0.441  | 0.686 | 0.798  | 0.796 | 0.550 |
| C03     | -0.116 | -0.118 | -0.073 | -0.078 | -0.004 | -0.008 | 0.453  | 0.426  | 0.588  | 0.706  | 0.780 | 0.892  | 0.862 | 0.900 |
| FO1     | 0.150  | 0.159  | 0.332  | 0.277  | 0.282  | 0.344  | 0.445  | 0.495  | 0.511  | 0.497  | 0.490 | 0.385  | 0.372 | 0.467 |
| G01     | 0.062  | 0.051  | 0.135  | 0.089  | 0.163  | 0.108  | 0.438  | 0.477  | 0.795  | 0.736  | 0.828 | 0.969  | 0.861 | 0.78  |
| K02     | -0.305 | -0.282 | -0.205 | -0.265 | -0.113 | -0.305 | 0.264  | 0.295  | 0.119  | 0.097  | 0.422 | 0.652  | 0.831 | 0.669 |
| K05     | -0.147 | 0.027  | 0.007  | -0.053 | 0.037  | 0.084  | 0.329  | 0.176  | 0.263  | 0.168  | 0.645 | 0.716  | 0.792 | 0.701 |
| K08     | 0.348  | 0.266  | 0.350  | 0.315  | 0.333  | 0.407  | 0.361  | 0.185  | 0.448  | 0.416  | 0.377 | 0.468  | 0.403 | 0.417 |
| K11     | -0.062 | -0.036 | -0.105 | -0.092 | 0.043  | -0.081 | 0.193  | 0.179  | 0.514  | 0.390  | 0.765 | 0.730  | 0.469 | 0.558 |
| Nano    |        |        |        |        |        |        |        |        |        |        |       |        |       |       |
| C02     | -0.479 | -0.260 | -0.208 | -0.409 | -0.297 | -0.345 | -0.050 | 0.144  | 0.173  | 0.151  | 0.249 | 0.333  | 0.330 | 0.264 |
| C03     | -0.275 | -0.261 | -0.211 | -0.257 | -0.080 | -0.206 | 0.113  | 0.031  | 0.247  | 0.192  | 0.363 | 0.355  | 0.288 | 0.256 |
| FO1     | -0.244 | -0.154 | -0.286 | -0.092 | -0.025 | 0.101  | 0.182  | 0.050  | 0.148  | 0.039  | 0.015 | 0.056  | 0.104 | 0.105 |
| G01     | -0.304 | -0.172 | -0.313 | -0.189 | -0.165 | -0.117 | -0.063 | -0.178 | 0.100  | 0.001  | 0.286 | 0.325  | 0.369 | 0.053 |
| K02     | -0.321 | -0.149 | -0.384 | -0.152 | 0.022  | 0.035  | 0.223  | 0.251  | -0.027 | -0.135 | 0.433 | 0.229  | 0.559 | 0.523 |
| K05     | -0.389 | -0.318 | -0.680 | -0.546 | -0.267 | -0.394 | -0.484 | -0.248 | -0.407 | -0.458 | 0.053 | -0.034 | 0.102 | 0.198 |
| K08     | 0.353  | 0.244  | 0.508  | 0.472  | 0.455  | 0.436  | 0.406  | 0.397  | 0.473  | 0.369  | 0.408 | 0.546  | 0.380 | 0.384 |
| K11     | -0.138 | -0.088 | -0.257 | -0.243 | -0.134 | -0.293 | 0.073  | 0.026  | 0.175  | 0.201  | 0.296 | 0.312  | 0.434 | 0.501 |
| Pico    |        |        |        |        |        |        |        |        |        |        |       |        |       |       |
| C02     | -0.383 | -0.188 | -0.186 | -0.199 | -0.119 | -0.162 | 0.188  | 0.143  | 0.162  | 0.241  | 0.257 | 0.291  | 0.377 | 0.205 |
| C03     | -0.202 | -0.258 | -0.259 | -0.282 | -0.143 | -0.160 | 0.017  | -0.019 | 0.148  | 0.191  | 0.194 | 0.248  | 0.230 | 0.300 |
| FO1     | -0.071 | -0.091 | -0.054 | -0.032 | 0.050  | 0.129  | 0.205  | 0.144  | 0.216  | 0.141  | 0.170 | 0.134  | 0.031 | 0.172 |
| G01     | 0.019  | -0.061 | 0.051  | -0.032 | 0.019  | 0.008  | 0.156  | 0.162  | 0.323  | 0.188  | 0.338 | 0.308  | 0.344 | 0.360 |
| K02     | -0.245 | -0.253 | -0.257 | -0.275 | -0.243 | -0.230 | -0.046 | 0.010  | -0.067 | -0.101 | 0.065 | -0.030 | 0.203 | 0.089 |
| K05     | -0.087 | 0.031  | 0.014  | -0.027 | 0.103  | 0.157  | 0.057  | 0.261  | 0.130  | 0.339  | 0.316 | 0.255  | 0.368 | 0.404 |
| K08     | 0.032  | 0.055  | -0.013 | 0.228  | 0.262  | 0.201  | 0.240  | 0.069  | 0.262  | 0.281  | 0.177 | 0.284  | 0.222 | 0.327 |
| K11     | -0.197 | -0.216 | -0.194 | -0.146 | -0.046 | -0.071 | -0.005 | 0.033  | 0.163  | 0.076  | 0.236 | 0.049  | 0.092 | 0.179 |

Table 3 Parameters derived from the dilution experiments (EXPb) in the ECS-Kuroshio.  $g_{max}$ : maximum growth rate (d-1). m: mortality rate by microzooplankton grazing (d-1).  $g_0$ : net growth rate measured in the non-enriched and non-diluted bottles (d-1).  $g_{en}$ : net growth rate measured in the enriched and non-diluted bottles (d-1).  $r^2$ : coefficient of determination defined from the linear regression of the apparent growth rate of total chlorophyll *a* concentrations against dilution factors. p: p-value. Pico: chlorophyll smaller than 2 µm. Nano: chlorophyll between 2 and 11 µm. Micro: chlorophyll larger than 11 µm. Total: total chlorophyll from pico- to micro.

448

| Station | Pico  |        |        |        | Nano  |       |        | Micro  |        |        |        | Total  |        |       |            |        |                |        |
|---------|-------|--------|--------|--------|-------|-------|--------|--------|--------|--------|--------|--------|--------|-------|------------|--------|----------------|--------|
|         | gmax  | m      | g.     | gen    | gmax  | m     | g.     | gen    | gmax   | m      | g.     | gen    | gmax   | m     | g ₀ | gen    | r 2 | р      |
| A05a    | 0.283 | 0.887  | 0.415  | 0.681  | 1.181 | 1.345 | -0.267 | 0.181  | 0.913  | 0.962  | 0.059  | 0.045  | 1.059  | 0.619 | 0.199      | 0.492  | 0.757          | < 0.0  |
| A05b    | 0.931 | 1.106  | -0.109 | 0.279  | 1.354 | 1.050 | -0.505 | -0.239 | 0.477  | 0.583  | -0.030 | 0.107  | 1.073  | 1.051 | -0.232     | 0.113  | 0.901          | < 0.01 |
| A05c    | 0.501 | 0.647  | -0.025 | 0.190  | 1.298 | 1.192 | -0.183 | -0.066 | 0.313  | 0.500  | -0.269 | 0.201  | 0.828  | 0.752 | -0.074     | 0.122  | 0.875          | < 0.01 |
| A06a    | 0.179 | 0.814  | 0.440  | 0.646  | 0.865 | 1.270 | 0.247  | 0.341  | 0.232  | 0.597  | -0.315 | 0.339  | 0.941  | 0.381 | 0.347      | 0.550  | 0.541          | < 0.01 |
| A06b    | 0.648 | -0.398 | -0.869 | -1.020 | 0.947 | 0.247 | -0.789 | -0.629 | -0.118 | -0.037 | -0.038 | 0.065  | -0.052 | 0.711 | -0.735     | -0.714 | 0.750          | < 0.01 |
| A08a    | 0.434 | 0.458  | -0.097 | 0.035  | 1.448 | 1.289 | -0.072 | -0.150 | 0.401  | 0.564  | -0.537 | 0.181  | 0.765  | 0.775 | -0.113     | 0.009  | 0.856          | < 0.01 |
| A08b    | 0.370 | 0.846  | -0.040 | 0.509  | 0.652 | 1.068 | -0.259 | 0.430  | 0.553  | 1.122  | -0.620 | 0.529  | 0.937  | 0.471 | -0.123     | 0.488  | 0.693          | < 0.01 |
| A09a    | 0.488 | 0.417  | -0.399 | -0.026 | 0.894 | 0.734 | -0.182 | -0.082 | 0.353  | 0.022  | -0.474 | -0.235 | 0.526  | 0.640 | -0.324     | -0.052 | 0.760          | <0.01  |

Table 4 Parameters derived from relationship of phytoplankton growth rates against logarithmically transformed concentrations of enriched nitrate in the gradient enrichment experiments (EXPa). Slope: sensitivity of phytoplankton growth rate to logarithmically transformed concentrations of enriched nitrate. Intercept: growth potential at the low nitrate concentration.  $r^2$ : coefficient of determination defined from the linear regression of growth rate of size-fractionated chlorophyll *a* concentrations against logarithmically transformed concentrations of enriched nitrate. Pico: chlorophyll smaller than 2 µm. Nano: chlorophyll between 2 and 11 µm. Micro: chlorophyll larger than 11 µm.

- 456
- 457

| Station _ |       | Pico      |                |        | Nano      | Micro          |       |           |       |
|-----------|-------|-----------|----------------|--------|-----------|----------------|-------|-----------|-------|
| -         | Slope | Intercept | r 2 | Slope  | Intercept | r 2 | Slope | Intercept | r²    |
| C02       | 0.281 | 0.178     | 0.848          | 0.370  | 0.131     | 0.831          | 0.458 | 0.492     | 0.846 |
| C03       | 0.295 | 0.121     | 0.922          | 0.308  | 0.177     | 0.830          | 0.560 | 0.611     | 0.914 |
| F01       | 0.074 | 0.129     | 0.317          | 0.120  | 0.067     | 0.420          | 0.077 | 0.430     | 0.368 |
| G01       | 0.203 | 0.243     | 0.866          | 0.272  | 0.085     | 0.688          | 0.448 | 0.657     | 0.817 |
| K02       | 0.213 | -0.014    | 0.883          | 0.364  | 0.233     | 0.726          | 0.531 | 0.353     | 0.872 |
| K05       | 0.188 | 0.251     | 0.772          | 0.355  | -0.165    | 0.729          | 0.419 | 0.439     | 0.843 |
| K08       | 0.070 | 0.231     | 0.242          | -0.038 | 0.426     | 0.213          | 0.045 | 0.386     | 0.162 |
| K11       | 0.167 | 0.077     | 0.750          | 0.394  | 0.201     | 0.943          | 0.403 | 0.409     | 0.744 |

---

## Author Response (AR2)

Reply to RC1

The revision and responses are largely satisfactory. I still feel that the writing (English and wording) needs to be improved.

> Thanks for suggestion. The manuscript has been edited carefully by two native-English-speaking professional editors from ELSS, Inc. (https://www.elss.co.jp/en). The edited words and phrases were everywhere and thus we do not highlight them in the revised manuscript. However, please note that these revisions are all for English writing.

Methods:

Please add a "Data analysis" section in M&M to explain the objectives and procedures of statistical analyses, for example the regression and ANOVA analyses in Figure 5-7, and others.

> Thanks for nice suggestion. We added the following description.

> 2.5 Data analysis

> To quantify the sensitivity of phytoplankton growth rates to nutrient supply rates, we calculated the slopes of linear regressions of growth rates for the size-fractionated chlorophyll a concentrations versus the logarithms of the enriched nitrate concentrations. We then computed the Pearson correlation coefficient of these slopes to nitrate + nitrite and phosphate concentrations and microzooplankton biomass at the beginning of each incubation. A one-way analysis of variance (ANOVA) with a post-hoc Tukey honestly significant difference test was used to compare maximum growth rates, mortality rates, and net growth rates among the three size fractions.

Line 131: I think you mean:

"In addition, for evaluating nutrient limitation on phytoplankton growth, extra triplicate non-diluted bottles (100%) for EXPb were conducted without nutrient amendment." Is my understanding correct?

> Yes, you are right. We revised the phrase as you mentioned as highlighted in yellow at line 135 to 136.

Line 173: Please explain how to calculate "95 percent confidence intervals of the vertical profiles".

> Thanks for comments. We described "95 percent confidence intervals obtained by a bootstrap process" in the Figure 2 caption.

Line 207: Should be "SI figure 2".

> Thanks for notice. Yes. The style should likely be Fig S2 based on the author instruction. We corrected it.

Discussion

The first paragraph seems just repeating the Introduction.

> No. We wanted to emphasize the topographic features are very specific at the Tokara Strait and important for nutrients supply in the surface layers. So, we keep this paragraph.